# Integrator is recruited to promoter-proximally paused RNA Pol II to generate *Caenorhabditis elegans* piRNA precursors

Toni Beltran[1,2,*,†], Elena Pahita[1,2], Subhanita Ghosh[1,2], Boris Lenhard[1,2] & Peter Sarkies[1,2,**]

## Abstract

Piwi-interacting RNAs (piRNAs) play key roles in germline development and genome defence in metazoans. In *C. elegans*, piRNAs are transcribed from > 15,000 discrete genomic loci by RNA polymerase II (Pol II), resulting in 28 nt short-capped piRNA precursors. Here, we investigate transcription termination at piRNA loci. We show that the Integrator complex, which terminates snRNA transcription, is recruited to piRNA loci. Moreover, we demonstrate that the catalytic activity of Integrator cleaves nascent capped piRNA precursors associated with promoter-proximal Pol II, resulting in termination of transcription. Loss of Integrator activity, however, does not result in transcriptional readthrough at the majority of piRNA loci. Taken together, our results draw new parallels between snRNA and piRNA biogenesis in nematodes and provide evidence of a role for the Integrator complex as a terminator of promoter-proximal RNA polymerase II during piRNA biogenesis.

**Keywords** integrator complex; non-coding RNA; Piwi-interacting RNAs (piRNAs); transcription termination

**Subject Categories** Chromatin, Transcription & Genomics; RNA Biology

**The EMBO Journal (2021) 40: e105564**

See also: **AC Berkyurek et al** (March 2021)

## Introduction

Piwi-interacting RNAs (piRNAs) are a class of small RNAs essential for germline development and transposable element silencing in metazoans (Ozata *et al*, 2019). piRNA biogenesis is best characterized in the fruit fly *Drosophila melanogaster* and in the nematode *Caenorhabditis elegans*. In *D. melanogaster*, piRNAs are transcribed from a small number of genomic clusters to produce > 10 kb long piRNA precursors. piRNA precursors are then processed by nuclease activities into 27–35 nucleotide long primary piRNAs. Upon detection of transposable element transcripts, piRNAs initiate a cycle of coupled cleavage reactions involving transposable element mRNAs and piRNA precursor RNAs. This coupling is known as the Ping-pong cycle, which allows amplification of piRNA populations while simultaneously degrading TE mRNAs (Brennecke *et al*, 2007; Gunawardane *et al*, 2007).

*Caenorhabditis elegans* piRNAs present remarkable differences to the *Drosophila* system. Mature piRNAs in *C. elegans* are 21 nucleotides long with a strong bias towards uracil as the first nucleotide (21U-RNAs; (Ruby *et al*, 2006; Batista *et al*, 2008; Das *et al*, 2008)). *Caenorhabditis elegans* piRNA loci are individual transcriptional units. The majority of piRNA loci localize to two ~3 Mb cluster regions on chromosome IV containing more than 15,000 loci and are demarcated by a conserved GTTTC motif at their promoter regions (Ruby *et al*, 2006; Billi *et al*, 2013); these loci are known as motif-dependent piRNAs or type I piRNAs. Another set of low-abundance piRNAs are generated from almost 10,000 loci distributed across the *C. elegans* genome and independently of an upstream motif; these are known as motif-independent piRNAs (Gu *et al*, 2012) or type II piRNAs. Both types of loci are transcribed as short 27-40 nt capped RNA precursors by RNA polymerase II (Pol II) (Gu *et al*, 2012). These short precursors have been proposed to result from pausing of Pol II at piRNA promoters (Gu *et al*, 2012; Beltran *et al*, 2019). We previously described a region of low melting temperature (high AT content) centred 33 nt downstream of piRNA TSSs across multiple nematode species, which may have a role in RNA polymerase II termination (Beltran *et al*, 2019). However, the exact mechanisms controlling this process are largely unknown. In particular, the role of Pol II-associated factors in promoting pausing and premature termination of Pol II remain unexplored.

Multiple lines of evidence suggest that the mechanisms of piRNA transcription in nematodes evolved by co-option of factors controlling the transcription of small nuclear RNAs (snRNAs). First, transcription of piRNAs requires the small nuclear RNA activating protein complex SNAPc (Kasper *et al*, 2014). SNAPc forms a distinct complex in the germline with the nematode-specific pseudokinase PRDE-1 (Kasper *et al*, 2014; Weick *et al*, 2014; Weng *et al*, 2019), which binds to the upstream Ruby motif

1 MRC London Institute of Medical Sciences, London, UK
2 Institute of Clinical Sciences, Imperial College London, London, UK
*Corresponding author. Tel: +34 93 316 01 00; E-mail: toni.beltran@crg.eu
**Corresponding author. Tel: +44 7570790430; E-mail: psarkies@imperial.ac.uk
†Present address: Centre for Genomic Regulation, Barcelona, Spain

directly (Weng *et al*, 2019). In addition, the GTTTC motif is found in snRNA promoters in basal nematodes (Beltran *et al*, 2019). Transport of piRNA precursors to their processing sites in perinuclear phase-separated P-granules is achieved by a specialized protein complex named PETISCO/PICS (Cordeiro Rodrigues *et al*, 2019; Zeng *et al*, 2019). Interestingly, components of this complex are required for the biogenesis of SL1 RNAs, the transsplicing leaders preceding the majority of *C.elegans* mRNAs, which are also related to snRNAs (Cordeiro Rodrigues *et al*, 2019). Altogether, these observations suggest that snRNAs and piRNAs share mechanisms of transcription and processing.

In metazoans, termination of snRNA genes requires the Integrator complex (Baillat *et al*, 2005). Integrator is a protein complex containing 12–14 subunits, which associates co-transcriptionally with the carboxy terminal domain (CTD) of Pol II. Two of the subunits of the complex, Ints-9 and Ints-11, are homologs of CPSF73, the catalytically active subunit of the cleavage and polyadenylation complex (CPSF) responsible for the formation of 3′ ends of protein-coding gene mRNAs. Ints-11 possesses the endonuclease activity responsible for cleavage of nascent snRNAs to generate snRNA 3' ends. This cleavage is coupled to Pol II termination.

While the best described function for Integrator is at snRNA loci, Integrator may also have functions in transcription of other Pol II-dependent transcripts. Integrator was shown to promote early termination of Pol II at the HIV LTR promoter, attenuating HIV transcription (Stadelmayer *et al*, 2014). A role for Integrator in conferring processivity to Pol II during the elongation stage of transcription has also been reported (Gardini *et al*, 2014; Stadelmayer *et al*, 2014). Additionally, Integrator mediates the biogenesis of enhancer RNAs (eRNAs) by promoting early termination of Pol II (Lai *et al*, 2015). Finally, Integrator has been proposed to terminate lncRNA transcription (Nojima *et al*, 2018).

Here, we discover that the Integrator complex is involved in piRNA biogenesis in nematodes. We show that Integrator activity is required for accumulation of mature piRNAs and their silencing activity. Integrator associates with piRNA clusters in germ cells, and its catalytic activity is essential to produce short-capped RNA precursors via co-transcriptional cleavage of nascent piRNA precursors. Our results provide new insights into the mechanisms of piRNA biogenesis in nematodes and uncover novel functions of the Integrator complex in the regulation of promoter-proximal Pol II pausing and the biogenesis of non-coding RNAs.

# Results

## Integrator is required for the biogenesis of short piRNA precursors

Prompted by the evolutionary relationship between piRNAs and snRNAs (Beltran *et al*, 2019), we sought to understand whether Integrator is involved in piRNA biogenesis in *C. elegans*. We performed RNAi against the catalytic subunit of the complex, *ints-11* and measured changes in piRNA abundance upon knock-down by sequencing small RNAs. We observed a decrease in total piRNA abundance in *ints-11*-depleted worms compared with the empty vector (EV) controls (Figs 1A and EV1A–C). These differences were also observed when comparing the piRNA abundance at each locus between EV and *ints-11* RNAi-treated animals (Fig 1B). Additionally, we assessed the silencing activity of piRNAs using a strain carrying a germline histone H2B::GFP transgene as a sensor of piRNA activity (Bagijn *et al*, 2012). Upon knock-down of *ints-11*, we observed an increase in the proportion of GFP-positive animals (Fig EV1D). Together these results suggested that Integrator functions in piRNA biogenesis.

To test whether Integrator acts at the transcriptional level, we performed high-throughput sequencing of short (18–36 nt) RNAs with a 5′ cap, allowing us to capture short-capped piRNA precursors. Upon knock-down of *ints-11*, we observed a decrease in piRNA precursors from piRNA loci with the characteristic GTTTC core (Ruby) motif (motif-dependent piRNAs; Figs 1C and D, and EV1E). We also observed a marked increase in motif-dependent piRNA precursor length upon Integrator depletion (Fig EV1E). To characterize the length distribution of piRNA precursors in greater detail, we performed high-throughput sequencing of capped RNAs up to a length of 75 nt. The previously described peak of ~28 nt piRNA precursors (Weick *et al*, 2014) was accompanied by an additional population of longer precursors up to ~75 nt long (Fig 1E and F). This population was rare in wild-type nematodes, but accumulated substantially upon *ints-11* depletion (Fig 1E and F). The increase in precursor length observed from the overall distributions was confirmed by comparing the difference in length at each locus between *ints-11* RNAi and EV controls (Fig EV1F and G).

Altogether, these data strongly suggest a role for Integrator in the termination of short motif-dependent piRNA precursor transcripts. Interestingly, we did not detect a decrease in abundance of motif-independent piRNA precursors, suggesting that their biogenesis may not depend on Integrator (Fig EV1H).

---

**Figure 1.  Integrator is required for motif-dependent piRNA precursor abundance and their short length.**

A   Average mature piRNA abundance relative to EV in EV and *ints-11* RNAi samples normalized using miRNA-derived size factors. Error bars represent the standard error of the mean (5 replicates).

B   Distributions of $\log_2$ fold change in piRNA abundance in *ints-11* RNAi-treated nematodes compared with empty vector-treated nematodes in 5 replicates. Boxplots show the interquartile range with a line at the median; the whiskers extend to the greatest point no more than 1.5 times the interquartile range.

C   Average piRNA precursor abundance relative to EV in EV and ints-11 RNAi samples, normalized to the total number of short-capped RNA reads mapping to annotated WormBase TSSs (Chen *et al*, 2013). Error bars represent the standard error of the mean (5 replicates).

D   Distributions of $\log_2$ fold change in piRNA precursor abundance in *ints-11* RNAi-treated nematodes compared to empty vector-treated nematodes in 5 replicates. Boxplots show the interquartile range with a line at the median; the whiskers extend to the greatest point no more than 1.5 times the interquartile range.

E, F   Distributions of piRNA precursor length in *ints-11* RNAi-treated nematodes compared with empty vector-treated nematodes, obtained from deeply sequencing one pair of libraries (inserts sequenced up to 75nt). Distributions of total sequences (E) and total read counts (F) are shown.

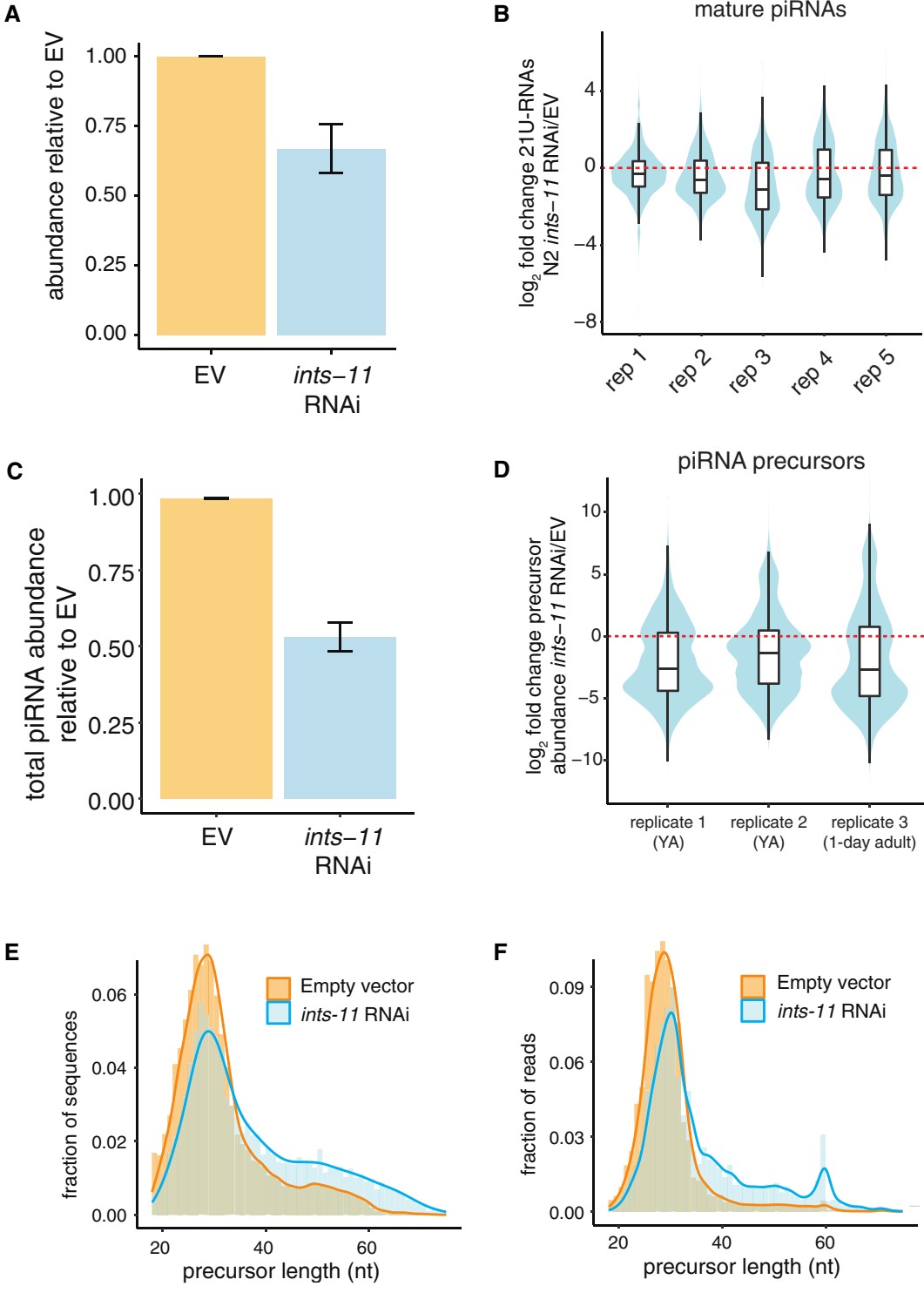

**Figure 1.**

**Integrator localizes to sites of piRNA biogenesis in germ cells**

We next investigated whether Integrator is recruited to sites of piRNA biogenesis. We used a strain where an endogenous deletion (*tm1616*) in the *ints-6* Integrator subunit, also known as *dic-1* in *C. elegans*, is rescued by an *ints-6*::3xFLAG::eGFP fusion transgene

(Gómez-Orte et. al., 2019). INTS-6::eGFP formed discrete foci in germ nuclei (Fig 2A), which start to form in the mitotic and transition zone regions of the germline and become most apparent in early-mid pachytene. To test whether these foci correspond to piRNA cluster regions, we crossed this strain with one carrying an mCherry-tagged version of the piRNA biogenesis factor PRDE-1,

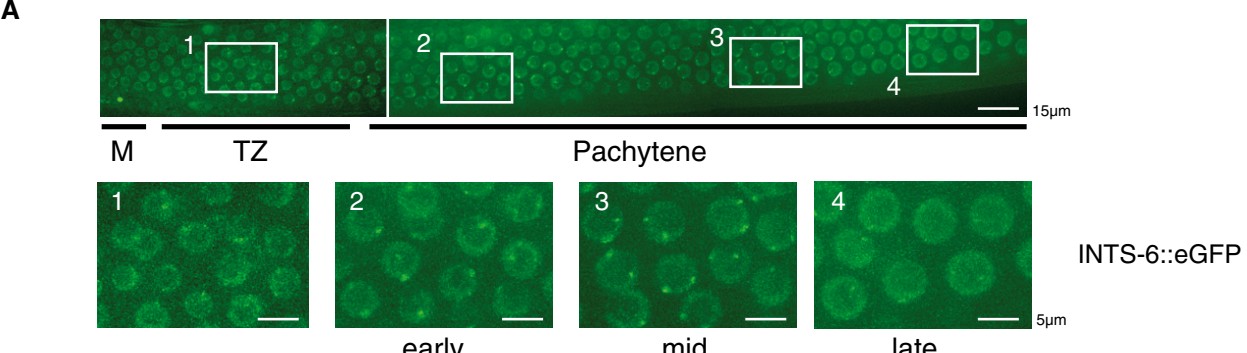

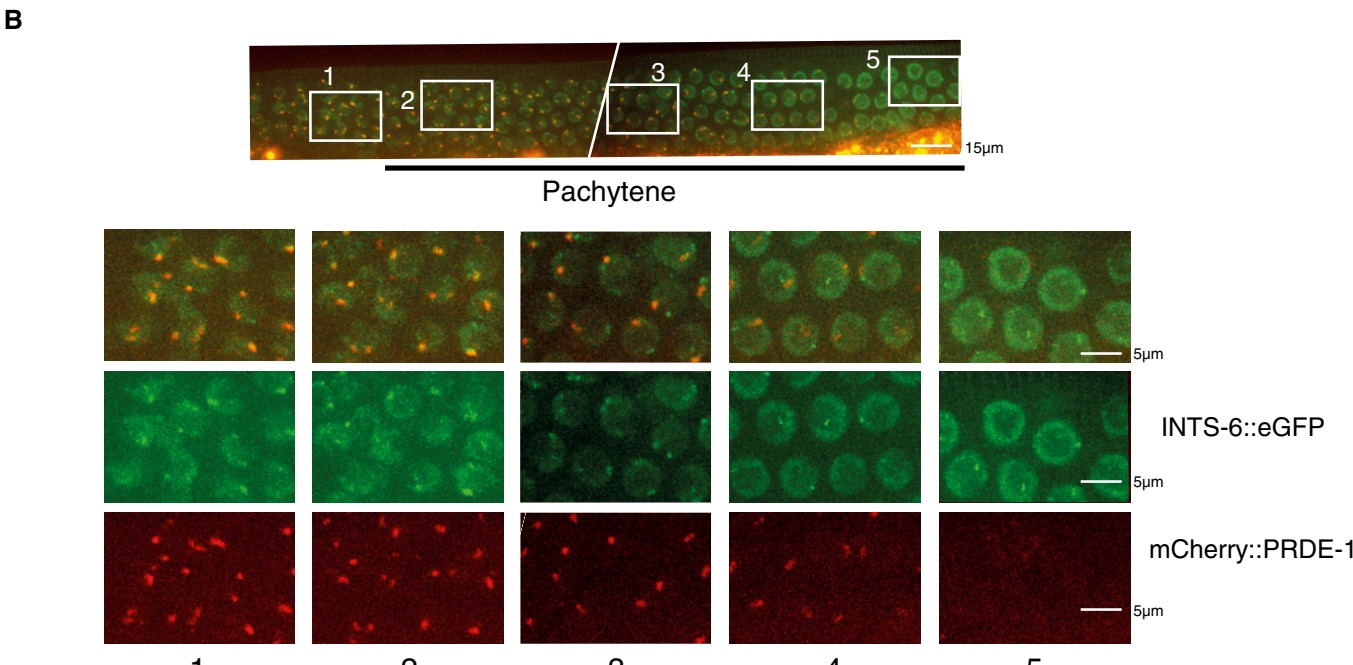

**Figure 2.  Integrator localizes to sites of piRNA biogenesis in germ cells.**

A   Live imagᵢng of a rescue transgene expressing an eGFP-tagged version of Integrator subunit 6 (INTS-6) in the germline of *C. elegans*. M = mitotic region,
TZ = transition zone.

B   Co-localization between INTS-6::eGFP and mCherry::PRDE-1 in germ cells. PRDE-1 and INTS-6 co-localize in early stages of meiosis (1,2) and up to mid-pachytene (3),
where an additional INTS-6 focus appears (3). When PRDE-1 foci are about to disappear, co-localization is lost (4), and the additional INTS-6 focus remains present
well after the loss of PRDE-1 foci (5).

which is known to bind to piRNA clusters (Weick *et al*, 2014; Weng *et al*, 2019). The majority of mCherry::PRDE-1 foci co-localized with INTS-6::eGFP foci across the germline (Fig 2B, quantification in Appendix Fig S1A). We observed a second population of INTS-6 foci appearing as cells progress into pachytene (Fig 2A), with mid-pachytene cells showing two clear INTS-6 foci, one of which co-localizes with PRDE-1 (Fig 2B, panels 2–3). Towards the end of pachytene, PRDE-1 foci became more diffuse and start to disappear, and INTS-6 signal in PRDE-1 regions was clearly reduced, resulting in no co-localizing foci (Fig 2B, panel 4). Interestingly, however, a single INTS-6 focus remained in late-pachytene cells well after the disappearance of PRDE-1 foci (Fig 2B, panel 5). In order to further

confirm these observations, we used CRISPR-mediated genome editing to tag the endogenous INTS-11 protein with an N-terminal FLAG tag and tested its localization by immunofluorescent staining. FLAG::INTS-11 also co-localized with PRDE-1 foci in germ cells (Appendix Fig S1B).

### Chromatin fractionation reveals two distinct populations of nascent piRNA precursors

We recently described a sequence signature of increased AT content downstream of motif-dependent 21U-RNAs resulting in a low melting temperature region (herefoth referred to as "termination

signal") (Beltran *et al*, 2019). We showed that the strength of this signal correlates with the length of motif-dependent piRNA precursors and suggested that it contributes to termination by promoting pausing of early elongating Pol II due to the low stability of the AT-rich nascent RNA-DNA hybrid. We also showed that the elongation factor TFIIS, which is involved in the rescue of backtracked Pol II complexes, is required for efficient motif-dependent piRNA biogenesis (Beltran *et al*, 2019). On the basis of these observations, we hypothesized that Integrator might function in motif-dependent piRNA biogenesis by processing nascent RNA associated with promoter-proximal Pol II, terminating transcription.

To test this model, we fractionated purified germ nuclei (Han *et al*, 2019) into chromatin bound and nucleoplasmic fractions and generated short-capped RNA libraries up to an insert size of 75 bp. This allowed us to distinguish chromatin-bound nascent piRNA precursor transcripts and nucleoplasmic precursors released from chromatin (Fig 3A). Importantly, as chromatin-bound precursors are predicted to be associated with transcribing Pol II, they report on the position of Pol II at piRNA loci, whereas nucleoplasmic precursors have already been released from the piRNA locus. Consistently, we observed longer precursors in the chromatin-bound fraction compared with nucleoplasm (Appendix Fig S2A and B).

In wild-type animals, the total length distribution of chromatin bound nascent RNA initiating at piRNA promoters followed a bimodal distribution, with a major peak at ~28 nt and a less abundant peak at ~48 nt. The ~48 nt peak was not present in nucleoplasmic RNA (Fig 3B). Weaker termination signals (lower AT content) downstream of the 21U-RNA locus were associated with a larger proportion of sequences mapping to the 48 nt peak (Fig 3C). Upon *ints-11* knock-down, we observed a decrease in nucleoplasmic and chromatin-bound precursor levels (Fig 3D), and an increase in length in both fractions (Fig 3E, Appendix Fig S3B). Mutation of *C. elegans* TFIIS (T24H10.1) led to a decrease in nucleoplasmic precursor levels and an increase in their length (Fig 3D and E, Appendix Fig S2B). However, chromatin-bound precursors had increased levels in *tfiis* mutants and showed an accumulation of nascent RNAs at the ~48 nt peak, suggesting that TFIIS activity is important for termination of Pol II at the 48 nt peak. Knock-down of *ints-11* in *tfiis* mutants led to a further increase in precursor length in both nucleoplasmic and nascent fractions (Fig 3E). The peak at 28 nt was also reduced relative to *tfiis* mutants, suggesting that the combined effect of loss of TFIIS and reduced Integrator activity interferes with initiation of transcription. This may result from torsional stress or spatial constraints resulting from the accumulation of Pol II at the 3′ end of the piRNA locus. Interestingly, the position of both nascent RNA peaks was shifted in *tfiis* mutants, from 27 to 29–30 nt and from 47-48 to 51 nt. These shifts were also seen when comparing ints-11 knock-down in wild-type and ints-11 knock-down in *tfiis* mutant animals (Fig 3E–G). These peak shifts are likely to result from reduced efficiency of cleavage of the 3′ end of nascent RNAs by Pol II in *tfiis* mutants after pausing and backtracking, suggesting that both peaks indeed correspond to Pol II pause sites on chromatin.

To support these observations at a locus-specific level, we calculated the mean length changes for all loci (Fig EV2A) and plotted the lengths of all detectable precursor sequences in each locus as a heat map (Fig EV2B). In addition, we plotted chromatin-bound precursor length heat maps across bins of increasing pausing

strength (Fig EV3). In EV controls, loci with weak pausing signals generally had both ~20–30 nt and ~40–50 nt long chromatin bound RNAs, indicating that both types of pausing events could occur at the same locus. However, at loci with strong pausing signals, a larger proportion of loci exhibited only ~20–30 nt chromatin-bound precursors. Similar trends were observed in *ints-11* RNAi-treated wild-type animals, and in *tfiis* mutants.

## Integrator cleaves nascent piRNA precursors associated with promoter-proximal Pol II

On the basis of our preceding observations, we hypothesized that Pol II pauses after it has transcribed ~28 nt, due to increased AT content found downstream of 21U-RNAs (Beltran *et al*, 2019). The paused polymerase could be resolved either through release from DNA, resulting in release of a ~28 nt piRNA precursor, or by Pol II resuming before arresting again ~48 nt downstream of the TSS. Since short ~28 nt nascent RNAs are contained within the polymerase, termination of Pol II at the ~28 nt peak to generate 28 nt precursors would be independent of Integrator. Termination at the ~48 nt peak, in contrast, would result from Integrator cleavage, as piRNA precursors at this stage are sufficiently long to be processed by the endonuclease activity of *ints-11* at the 5′ side of paused Pol II. Thus, we hypothesized that termination of the 48 nt peak would be dependent on the Integrator complex. This model would account for the increased proportion of ~48 nt nascent RNAs upon Integrator knock-down, and the small but consistent 3′ shift in the position of the two peaks in the absence of TFIIS.

To test this model, we searched for cleavage products with a 5′ monophosphate that might result from Integrator cleavage of nascent RNA (Fig 4A). We observed a population of 5′ monophosphate fragments with 5′ ends peaking at +38 nt of piRNA TSSs, which we designated as putative cleavage products (Fig 4B). *ints-11* knock-down resulted in a clear decrease in cleavage products (Fig 4B and C, Appendix Fig S3A and B), suggesting that these result from Integrator cleavage. These fragments had a median length of 20 nt (Fig 4D). Further, the 3′ position of cleavage fragments was centred 58 nt downstream of piRNA TSSs (Appendix Fig S3C). This implies that at the point at which cleavage occurs, Pol II has transcribed approximately 10 nt past the +48 pause site. The lack of a defined +58 peak of chromatin-bound RNAs (Fig 3E) implies that Pol II is rapidly terminated at this point. The *tfiis* mutation alone also resulted in a reduction in Integrator cleavage products (Fig 4B and C, Appendix Fg S3A and B). This indicates that TFIIS activity is important for Integrator-mediated processing of nascent RNAs, by promoting Pol II transcription through the +48 pause site. This model is consistent with the accumulation of ~48 nt nascent precursors in *tfiis* mutants (Fig 3D and E). Similarly, *ints-11* knock-down in a *tfiis* background resulted in a reduction in cleavage products (Fig 4B and C, Appendix Fig S3A and B).

An important test of our model is that stronger AT-rich signals would result in reduced reliance on Integrator cleavage due to a larger proportion of termination events happening at the 28 nt peak. In support of this, loci with stronger AT-rich signals tended to have reduced abundance of cleavage products (Fig 4E, Appendix Fig S3D). These data suggest that the AT-rich signal affects pausing and progression of Pol II to the 48 nt peak, and not Integrator processing efficiency. Consistently, the differences in

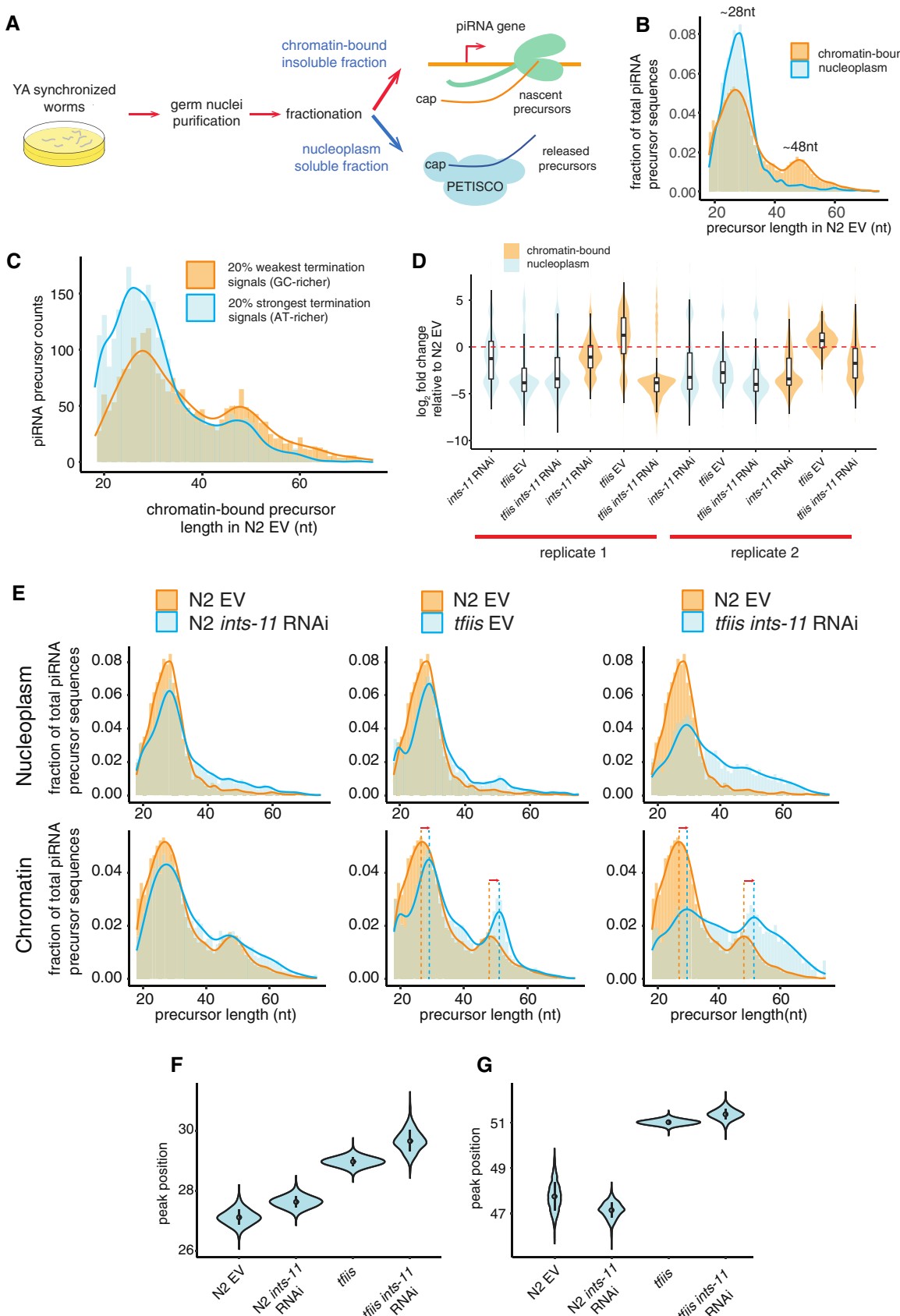

**Figure 3.**

**Figure 3. Chromatin fractionation identifies nascent piRNA precursors.**

A   Experimental design. Germ nuclei isolated from empty vector or *ints-11* RNAi-treated nematodes were fractionated into chromatin-bound and nucleoplasmic fractions. RNA corresponding to each fraction was purified and used as input for short-capped RNA libraries.

B   Length distribution of chromatin-bound and nucleoplasmic piRNA precursors in wild-type N2 nematodes, calculated from two merged replicates.

C   Length distribution of piRNA precursors generated from loci with weak and strong AT content enrichment downstream of the 21U-RNA in wild-type nematodes, calculated from two merged replicates.

D   Distributions of log2 fold change in piRNA precursor abundance in nucleoplasmic and chromatin fractions relative to N2 empty vector controls, for N2 *ints-11* RNAi, *tfiis* empty vector, and *tfiis ints-11* RNAi nematodes in two replicates. Significance for each distribution to be different from no change was tested using a two-tailed paired Wilcoxon rank-sum test, with all p-values < 2.2e-16. Boxplots show the interquartile range with a line at the median; the whiskers extend to the greatest point no more than 1.5 times the interquartile range.

E   Length distributions of nucleoplasmic and chromatin-bound piRNA precursors in N2 empty vector, N2 *ints-11* RNAi, *tfiis* empty vector and *tfiis ints-11* RNAi nematodes. Length shifts in nascent RNA in *tfiis* mutants are indicated with dashed vertical lines and arrows. The length distributions were calculated from two merged replicates (see Appendix Fig S2 for the length distributions in each replicate).

F, G   Distribution of the position of the ~28 nt (F) and ~48 nt (G) nascent RNA peaks in 2,000 bootstrapped subsamples of 3,000 precursors. Sampling probability was weighted by precursor abundance.

nascent piRNA precursor length between loci with strong and weak termination signals were still present upon knock-down of *ints-11*, suggesting that *ints-11* does not mediate the effect of the termination signal (Appendix Fig S4).

We wondered whether we could find any evidence of additional trimming of Integrator-released precursors. To test this, we categorized loci in bins according to the median length of their chromatin-bound precursors and examined the distributions of nucleoplasmic precursor length across these bins (Appendix Fig S5). This analysis showed that even loci with a large fraction of long chromatin-bound precursors (> 38 nt) produce nucleoplasmic precursors with a median length < 30 nt (Appendix Fig S5).This implies that RNA released from chromatin upon Integrator cleavage may be further trimmed to 28 nt in the nucleoplasm before further maturation to mature piRNAs.

## Pol II elongation control is independent of Integrator activity at the majority of piRNA loci

Having established that Integrator cleaves nascent piRNA precursors associated with paused Pol II, we set out to test whether Integrator acts to prevent further Pol II elongation beyond the second pause site at 48 nt downstream of the piRNA TSS. To investigate this, we performed super low-input carrier CAGE (Cvetesic *et al*, 2018) using chromatin-bound RNA as input material and subjected the libraries to paired-end sequencing. This approach captures 5′ ends of nascent long (>200 nt) capped RNAs, allowing us to identify putative piRNA precursor initiating 2 nt upstream of 21U-RNAs. Paired-end sequencing allowed us to obtain lower-bound estimates of their length from the position of the 3′ end of the fragments generated through random priming.

Nascent RNA libraries had much higher intron retention rates than a total RNA control (Fig EV4A and B), validating that chromatin fractionation captures nascent RNAs. Across samples, we observed an enrichment of unique CAGE tags initiating 2 nt upstream of motif-dependent 21U-RNAs corresponding to 659 loci in total (3.64% of Ruby motif-dependent piRNAs, Figs 5A and EV4C). These transcripts were found at low levels in N2 EV controls, as well as in *tfiis* mutants; however, *ints-11* knock-down led to an increase in their abundance (Fig 5B and C), both in an N2 background (2.83 median fold increase), and a *tfiis* background (22.62 median fold increase) (Fig 5B). Importantly, this

effect was highly specific for CAGE reads initiating 2 nt upstream of piRNAs (Fig 5C), confirming that this signal captures transcription from piRNA promoters. These transcripts peaked at a length of ~400–500 nt (Fig 5D), similarly to the library insert size, suggesting that this is a lower-bound estimate of their length. Thus, *ints*-11 knock-down leads to an increase in readthrough transcription at piRNA loci.

Long capped RNAs initiating 2 nt upstream of motif-independent piRNAs were much more abundant, with 4,437 motif-independent piRNA loci producing detectable long capped RNAs (45% of motif-independent loci) (Figs 5E and EV4D and E). Their abundance, however, was slightly reduced upon Integrator knock-down, in contrast to motif-dependent piRNAs (Fig 5F).

Despite the clear increase in readthrough transcription at motif-dependent piRNA loci upon *ints-11* knock-down relative to WT, we still only detected such events at a minority of loci (3.64%). This observation is not likely to result from low sensitivity of CAGE since the levels of short piRNA precursors are not significantly different between CAGE-detected and undetected piRNAs (Fig EV4F); and the number of detected motif-dependent piRNAs is saturated at the sequencing depth we obtained for our CAGE libraries (Fig EV4G). In order to better understand the factors influencing Pol II promoter escape at piRNA loci, we examined chromatin accessibility (Jänes *et al*, 2018) and germline H3K27ac levels (Han *et al*, 2019) at piRNA promoters. Motif-dependent piRNA promoters showed extremely low accessibility levels below genome-wide average (Fig EV5A); however, a clear increase in ATAC signal is observed around piRNA TSSs when comparing wild-type adult worms to *glp-1* mutants lacking a germline (1.5-fold mean fold increase) (Fig EV5B). Motif-independent piRNAs, in contrast, showed on average 10 times higher accessibility than genome-wide average, both in wild-type and *glp-1* nematodes (Fig EV5A and B). Both types of piRNA loci also showed clear differences in the levels of H3K27ac: motif-dependent loci showed extremely low levels, while motif-independent loci were highly enriched; this enrichment was especially high in the germline (Fig EV5C). The subset of motif-dependent piRNA promoters that were detected by CAGE tended to have slightly higher levels of H3K27ac and accessibility, but these are still very low relative to motif-independent loci (Fig EV5D and E). We thus conclude that the differences in read through transcription between motif-dependent and motif-independent piRNA promoters correlate with chromatin accessibility and H3K27ac status.

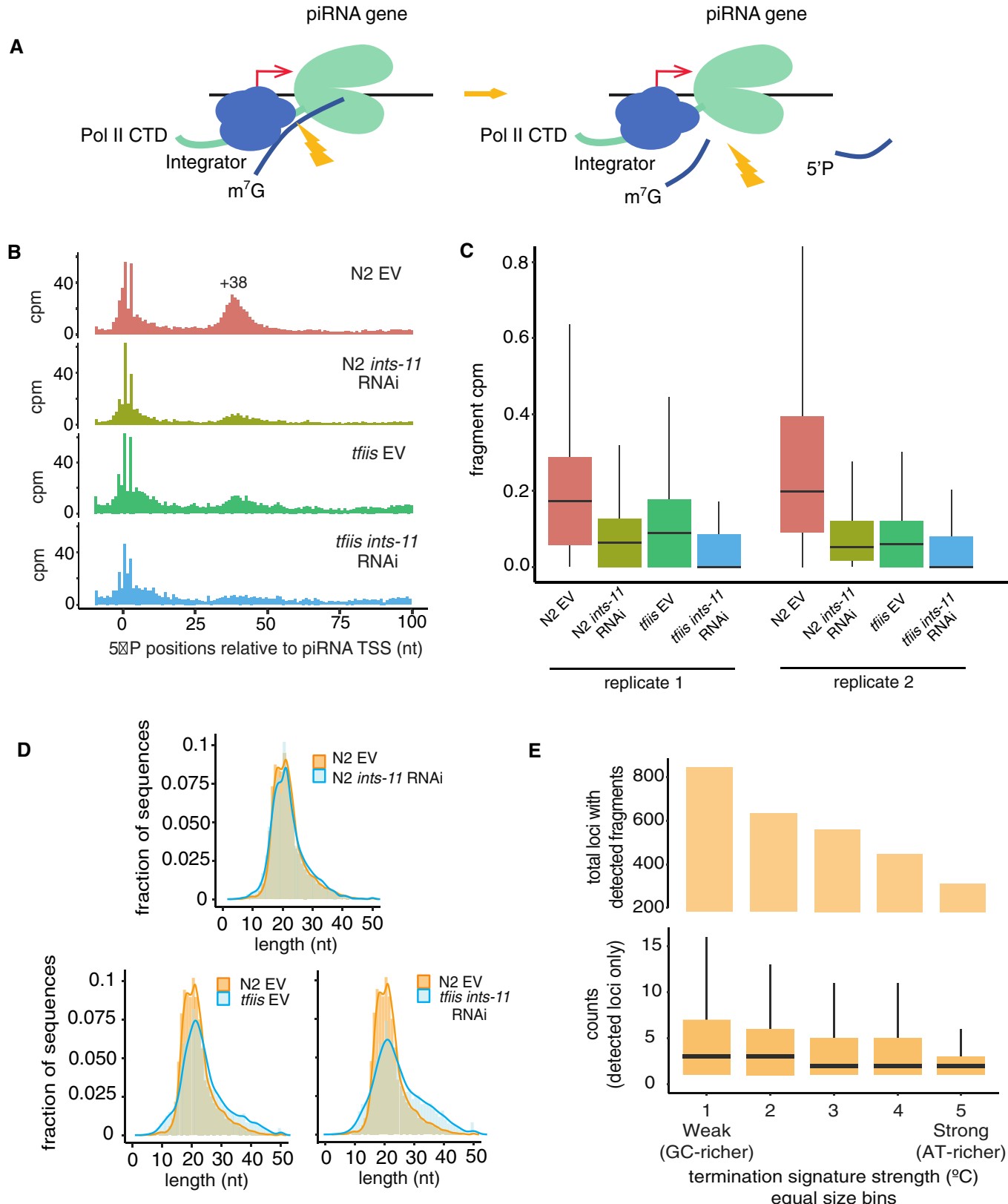

Figure 4.

**Figure 4.** **Integrator-mediated cleavage of nascent piRNA precursors generates short-capped RNAs.**

A Model for Integrator cleavage of nascent piRNA precursors associated with promoter-proximal Pol II. Cleavage results in the production of a short-capped RNA, and a 3′ degradation fragment.

B Signal of 5′ P small RNA 5′ ends mapping to piRNA loci as a function of the distance to piRNA TSSs, after removal of reads corresponding to mature 21U-RNAs (see Materials and Methods). The signal is normalized to counts per million of non-structural mapped reads. A peak of 5′ P ends centred at +38 from piRNA TSSs is observed.

C Distributions of counts per million of 5′ P ends mapping from +25 to +50 of piRNA TSSs across the different genotypes and conditions. Two independent biological replicates are shown. Boxplots show the interquartile range with a line at the median; the whiskers extend to the greatest point no more than 1.5 times the interquartile range.

D Length distribution of putative 3′ degradation fragments with 5′ P ends mapping at +28-+58 of the 5′ U of annotated 21U-RNAs.

E Number of loci with detected cleavage fragments in N2 empty vector nematodes across percentile bins of increasing termination signal strength (top panel). Distributions of fragment read counts per locus for detected loci across percentile bins of increasing termination signal strength (bottom panel). Boxplots show the interquartile range with a line at the median; the whiskers extend to the greatest point no more than 1.5 times the interquartile range. The data shown here correspond to one of two N2 EV replicates, see Appendix Fig S6 for this analysis across replicates and genotypes.

In contrast to piRNA loci *ints*-11 knock-down led to a dramatic increase in CAGE signal at all snRNA loci (Appendix Fig S6A and B), reflecting extensive readthrough past snRNA termination site (Fig 6). This is in accordance with previous analyses of the effect of *ints-11* knock-down on sRNA transcription, which has been described to result in fusion transcripts with downstream protein-coding genes (Gómez-Orte *et al*, 2019).

## Integrator attenuates protein-coding gene expression in *C. elegans*

We assessed the role of Integrator in protein-coding gene transcription by examining the changes in short-capped RNA and CAGE reads initiating at gene promoters (Chen *et al*, 2013). We identified a total of 437 genes with changes in short-capped RNA signal (FDR < 0.1; 371 upregulated, 66 downregulated, Appendix Fig S6C), as well as 470 genes with changes in CAGE signal (FDR < 0.1; 330 upregulated, 140 downregulated, Appendix Fig S11D). Changes in short and long-capped RNAs tended to correlate well, with very few genes showing changes in the opposite direction (Appendix Fig S6E–G). Interestingly, Integrator depletion also led to increased transcription of a subset of transposable elements from the Tc1 family (Appendix Fig S7A and B). The enrichment of upregulated genes suggests that Integrator-mediated termination negatively regulates transcription at certain genomic regions, potentially by cleaving upstream of paused polymerase to terminate transcription as has been recently reported (Elrod *et al*, 2019). Since motif-independent piRNA loci tend to localize within protein-coding gene promoters genome-wide, we wondered whether they may be generated as a by-product of premature Pol II termination at protein-coding genes by Integrator. Motif-independent piRNA loci, however, were not enriched in the promoters of genes that become upregulated upon Integrator knock-down (Appendix Fig S6H), consistent with an Integrator-independent biogenesis pathway for motif-independent piRNAs.

## Discussion

In this work, we gain insight on the fundamental mechanism of piRNA biogenesis in *C. elegans* and explore the role of the Integrator complex in the termination of promoter-proximal Pol II at piRNA loci.

### Parallels between snRNA and piRNA transcription

Multiple lines of evidence suggest that piRNA promoters evolved from snRNA promoters (preprint: Weng *et al*, 2018; Beltran *et al*, 2019; Cordeiro Rodrigues *et al*, 2019). Here, we explore the parallels between the mechanisms of piRNA and snRNA transcription. We show that the Integrator complex is recruited to sites of piRNA biogenesis upon piRNA transcription activation. It is unclear whether Integrator recruitment to piRNA promoters is directly mediated by the USTC complex (SNAPc/PRDE-1/TOFU-5), or whether Integrator is recruited to the initiating Pol II independently of SNAPc. Integrator has been shown to associate with the CTD of Pol II phosphorylated on Ser7 (Egloff *et al*, 2007, 2010). It will be of interest to explore whether Ser7-P is associated with piRNA biogenesis, and the mechanisms directing this modification. The presence of piRNA-independent Integrator foci may reflect Integrator-dependent regulation of snRNA genes. Indeed, snRNA genes have been shown to spatially interact and coalesce into Cajal bodies (Frey & Matera, 2001; Wang *et al*, 2016). This phenomenon may share some mechanistic aspects with the formation of piRNA foci in the *C. elegans* germline.

### Two distinct populations of piRNA precursors

By isolating chromatin-bound RNA from germ nuclei, we characterize two distinct populations of nascent piRNA precursors; one peaking at ~28 nt, and a second population of ~48 nt long, representing two Pol II pause sites on chromatin. In addition, we provide evidence that Integrator acts to cleave 53–63 nt nascent precursors past the +48 pause site, resulting in the release of 33–43 nt short-capped piRNA precursor transcripts (Fig 6). Several studies had set out to address the length of piRNA precursors in the past. Cecere *et al*, 2012 reported the existence of a ~75 nt long precursor detected by RACE, but high-throughput sequencing-based approaches detected 26–28 nt precursors (Gu *et al*, 2012; Weick *et al*, 2014). Our discovery of a second population of 40–60 nt nascent precursors reconciles these previous observations by suggesting that longer precursor transcripts, while rare, can be detected at piRNA loci.

Interestingly, a large fraction of nascent precursors peak at a length of ~28 nt, as observed for nucleoplasmic precursors. This raises the question of whether these are truly chromatin-bound. We previously suggested that increased AT content downstream of 21U-RNAs leads to Pol II pausing ~28 nt downstream of the TSSs

(Beltran *et al*, 2019). Consistent with this model, TFIIS mutation results in a 2–3 nt shift in both +28 and +48 peaks, suggesting pausing and backtracking of Pol II at both sites (Fig 6). A recent study also supports the existence of two Pol II pause sites at protein-coding genes in human cell lines (Aoi *et al*, 2020). Interestingly, accumulation of Pol II at the second pause site is observed upon NELF depletion, and coincides with a positioned downstream nucleosome (Aoi *et al*, 2020). *Caenorhabditis elegans* does not have NELF

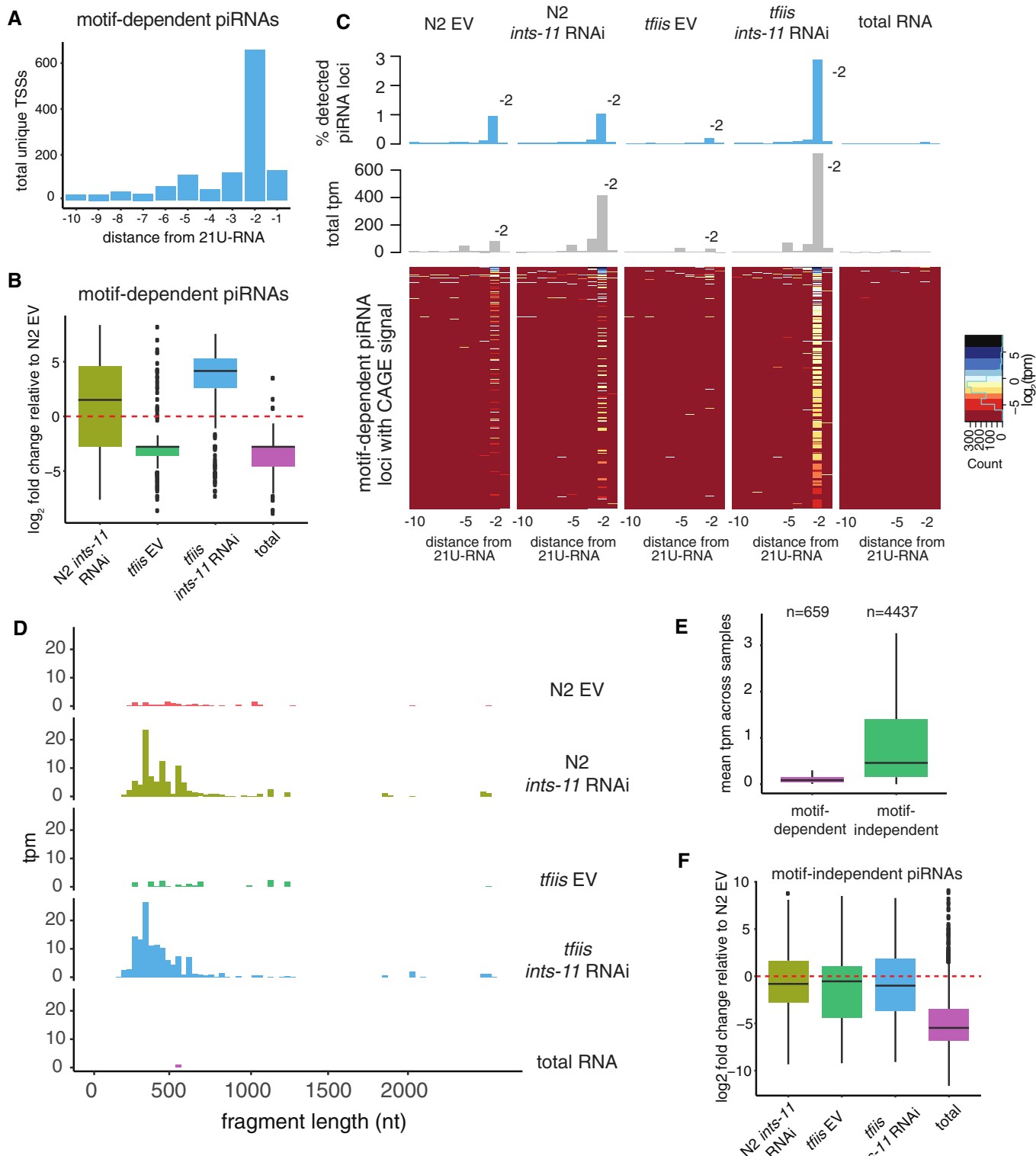

Figure 5.

**Figure 5.   Integrator knock-down results in transcriptional readthrough in a subset of motif-dependent piRNA loci.**

A   Number of unique CAGE-supported TSSs mapping to a 10 nucleotide region upstream of motif-dependent 21U-RNAs. A clear enrichment of TSS 2 nt upstream of 21U-RNAs is observed.

B   Distributions of $\log_2$ fold change in CAGE signal corresponding to 2 nt upstream motif-dependent piRNAs upon *ints*-11 RNAi knock-down in an N2 and a *tfiis* mutant background calculated from two merged replicates. N2 empty vector was used as a baseline for fold change calculation. Boxplots show the interquartile range with a line at the median; the whiskers extend to the greatest point no more than 1.5 times the interquartile range.

C   Heat map showing changes in CAGE signal upstream of motif-dependent 21U-RNAs upon *ints*-11 knock-down in an N2 and a *tfiis* mutant background. The total cpms corresponding to each nucleotide position upstream to 21U-RNAs are shown as a bar plot (top part).

D   Length distributions of CAGE fragments initiating 2 nt upstream of motif-dependent 21U-RNAs. The total counts in each length bin are normalized to counts per million of mapped reads.

E   Average normalized CAGE signal (transcripts per million) 2 nt upstream of motif-dependent and motif-independent piRNAs (excluding undetected loci). Boxplots show the interquartile range with a line at the median; the whiskers extend to the greatest point no more than 1.5 times the interquartile range.

F   Distributions of $\log_2$ fold change in CAGE signal corresponding to 2 nt upstream motif-independent piRNAs upon *ints*-11 RNAi knock-down in an N2 and a *tfiis* mutant background. N2 empty vector was used as a baseline for fold change calculation. Boxplots show the interquartile range with a line at the median; the whiskers extend to the greatest point no more than 1.5 times the interquartile range.

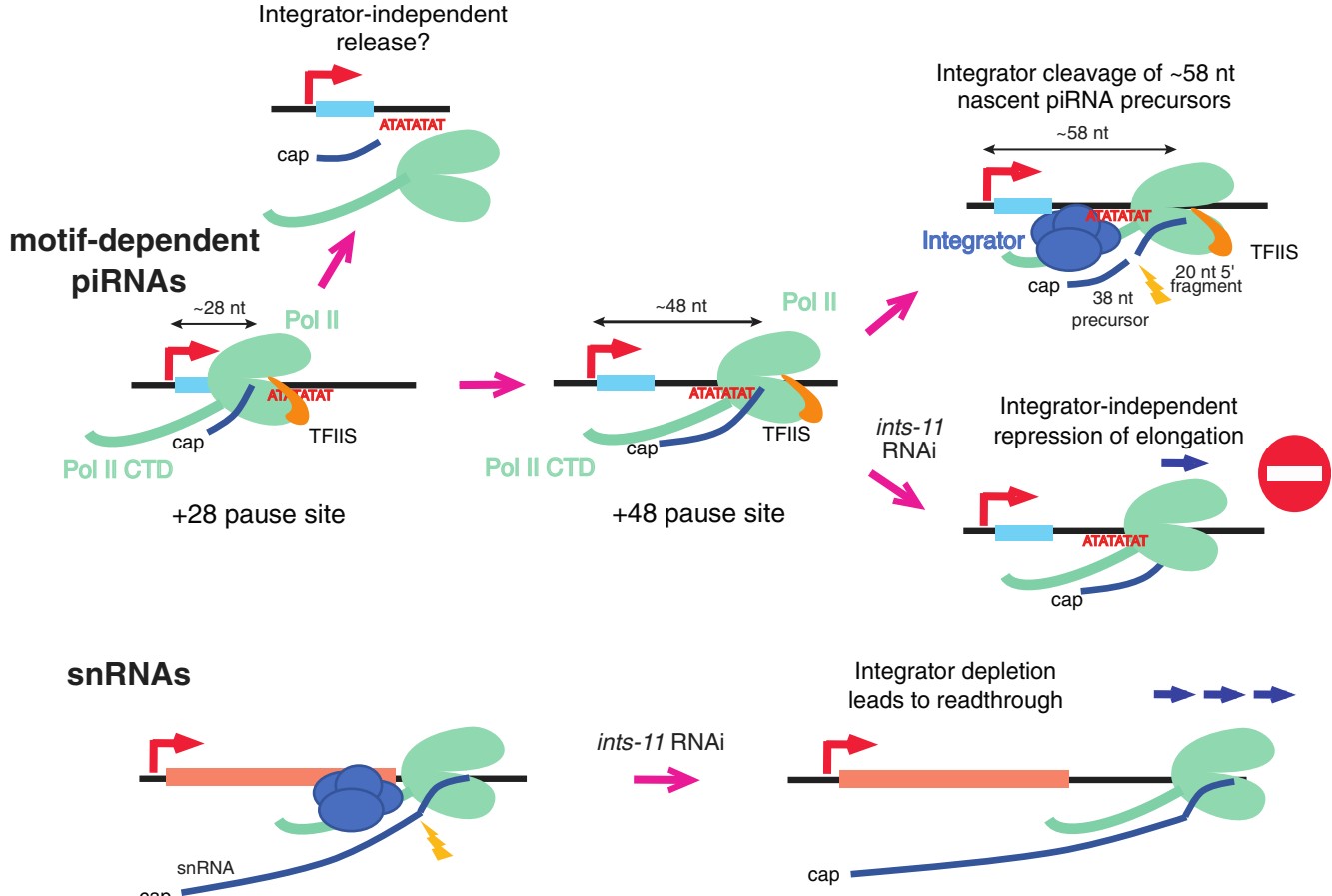

**Figure 6.   Model for the role of Integrator in piRNA transcription termination.**

At snRNA loci, Integrator generates snRNA 3′ ends leading to Pol II termination. *ints*-11 knock-down results in readthrough past termination sites and polyadenylation of snRNA transcripts. At motif-dependent piRNA loci, Pol II pauses ~ 28 nt and ~ 48 nt downstream of piRNA TSSs. ~28 nt short-capped RNAs may be generated via direct dissociation of Pol II from chromatin at the ~ 28 nt pause site. When Pol II transcribes past the ~ 48 nt pause site, Integrator acts to cleave nascent precursors approximately when Pol II is at + 58 from piRNA TSSs, generating ~ 38 nt short-capped RNAs and ~ 20 nt degradation fragments. *ints*-11 knock-down does not lead to Pol II readthrough, suggesting that additional mechanisms prevent Pol II elongation at piRNA loci.

(Maxwell *et al*, 2014) potentially favouring the transition of Pol II to the +48 peak. In addition, integrator cleavage fragments are reduced in loci with high AT content downstream of 21U-RNAs, suggesting that Pol II is less likely to reach the +48 peak in these loci. Pol II may be released from the +28 pause site independently of Integrator, and Integrator may act as a failsafe termination mechanism when

transcription continues beyond +48 (Fig 6). As indicated by the 5′ ends of cleavage fragments, Integrator-mediated termination releases ~38 nt capped RNAs, and these intermediates are likely further trimmed on their 3′ ends to generate the main class of ~28 nt nucleoplasmic precursors. Although we cannot formally rule out the possibility that Integrator promotes accumulation of Pol II at the +28 site, we favour the model outlined above given the known functions of Integrator in nascent RNA cleavage. It is also possible that Integrator depletion affects overall transcription initiation or nascent piRNA precursor stability, given the decrease in the levels of chromatin-bound precursors observed upon *ints-11* RNAi.

### Integrator and Pol II elongation control

Our analysis of the role for Integrator at piRNA loci in *C. elegans* fits with a recent study demonstrating that Integrator terminates promoter-proximal Pol II at protein-coding gene promoters to attenuate gene expression (Elrod *et al*, 2019). The short-capped RNAs produced as a by-product of Integrator cleavage at protein-coding gene promoters are rapidly degraded by the nuclear exosome. At *C. elegans* piRNA loci, instead, the resulting short-capped RNAs are handed in to a specialized RNA processing complex resulting in mature piRNA production (Cordeiro Rodrigues *et al*, 2019; Zeng *et al*, 2019). How piRNA precursors are recognized and channelled into the downstream processing pathways while avoiding nuclear RNA surveillance will be interesting to explore in the future.

Our data reveal an interesting interplay between TFIIS and Integrator function in the regulation of promoter-proximal Pol II. TFIIS mutation results in increased nascent precursors, and an accumulation of ~48 nt precursors, but a decrease in degradation fragments, suggesting that TFIIS activity modulates the ability of Integrator to process nascent RNAs. The observation that Integrator processing occurs downstream of the +48 pause site suggests that the role of TFIIS is to promote elongation through the +48 site prior to Integrator termination. Interestingly, *C. elegans* mutants of the RPB-9 subunit of Pol II exhibit defects in piRNA biogenesis similar to *tfiis* mutants (Berkyurek et al, cosubmitted). These data are consistent with previous studies indicating that TFIIS binding to Pol II is compromised in rpb-9Δ mutants in yeast (Sigurdsson *et al*, 2010). Indeed, the capacity of TFIIS to promote Pol II transcription through arrest sites *in vitro* depends on RPB-9 (Awrey *et al*, 1997). Understanding how the regulation of Pol II early elongation influences the ability of Integrator to prematurely terminate transcription will be an interesting avenue for research in the Pol II pausing field.

At snRNA loci, Integrator is required to terminate transcription and in its absence substantial readthrough transcription is observed (Baillat *et al*, 2005; Gómez-Orte *et al*, 2019). In our study, we observed similar evidence of readthrough transcription after Integrator knock-down at snRNA loci. In contrast, Integrator depletion did not result in readthrough transcription at most motif-dependent piRNA loci. This indicates that the cleavage activity of Integrator is not required to prevent elongation at motif-dependent piRNAs. Pol II elongation control is thus likely to operate independently of Integrator via additional mechanisms (Fig 6). One possibility is that elongation repression is chromatin-based; since motif-dependent piRNAs localize to broad domains of H3K27me3 (Beltran *et al*, 2019), and very low H3K27ac (Fig EV5D) and accessibility (Fig EV5A–C), a situation reminiscent of poised enhancers

(Creyghton *et al*, 2010). In contrast, motif-independent piRNAs, which show extensive readthrough, are in highly accessible H3K27ac-rich genomic regions. The tight control of Pol II elongation at motif-dependent piRNA loci may be a consequence of evolutionary pressure to avoid transcriptional interference between neighbouring piRNA loci given the high density of loci in piRNA cluster regions that are simultaneously activated.

### Integrator: piRNA functions beyond nematodes?

The involvement of Integrator in piRNA biogenesis adds to the existing evidence that snRNA and nematode piRNA transcription are functionally and evolutionarily related. Whether this principle extends to piRNAs in other organisms remains unknown; however, the piRNA biogenesis factor cut-off has been shown to bind to snRNA promoters in *Drosophila melanogaster* (Pritykin *et al*, 2017). Interestingly, Integrator was identified in genetic screens to be required for piRNA-mediated silencing of a reporter in the *Drosophila* germline (Czech *et al*, 2013) and in ovarian somatic cells (Handler *et al*, 2013), making Integrator a possible candidate to assist in termination of piRNAs in *Drosophila* and mouse. Our data add to the growing evidence indicating co-option of functional modules from a range of existing transcriptional and RNA metabolic processes in metazoan piRNA biogenesis, and expand the repertoire of non-coding RNAs under Integrator control in metazoans.

## Materials and Methods

### Nematode culture and RNAi

*C. elegans* nematodes were grown at 20°C in standard nematode growth medium (NGM) agar plates feeding on OP50 *E. coli*. RNAi clones from the Ahringer library (Kamath & Ahringer, 2003) were grown overnight in LB supplemented with 50 μg/ml ampicillin and 25 μg/ml tetracycline. The next morning, cultures were diluted in LB 50 μg/ml ampicillin, grown to an OD of 0.5–0.6 and seeded into 1-week-old NGM plates supplemented with 1 mM IPTG and 50 μg/ml ampicillin. Plates were dried for 48 h before plating synchronized L1 worms obtained by hypochlorite treatment followed by 24 h starvation in M9 media. See Appendix Table S1 for nematode strains used in this study, and Appendix Table S2 for oligonucleotides used in this study.

### Live imaging

Live imaging of SX1316 piRNA sensor worms, as well as INTS-6::eGFP::3xFLAG and PRDE-1::mCherry transgenes was carried out in a DeltaVision fluorescence microscope system. Worms were picked into a 2 μl drop of M9 with 0.5 μM levamisole on a slide, and subsequently immobilized in a hydrogel matrix with microbeads as described in Dong *et al*, 2018. Live imaging was carried out within 15 min of slide preparation.

### Immunofluorescent staining

Mouse anti-FLAG M2 monoclonal antibody (F1804, Sigma) preadsorbed with *C. elegans* lysates to remove background was used for

immunofluorescent staining. 15–20 worms were picked onto 15 µl M9 Tween 0.1% on a poly-lysine-coated slide and dissected using a gauge needle. Another 15 µl 2% paraformaldehyde in M9 Tween 0.1% were added, some solution was removed and a coverslip was placed on top to let the germlines fix for 5 minutes. The slides were then frozen in liquid nitrogen, worms were cracked by quickly removing the coverslip, and slides were fixed in −20°C methanol for at least 1 minute. Slides were then washed by immersion in PBST for 5 min, 3 times, and blocked with PBST with 1% BSA for 1 h at room temperature. Slides were then incubated with mouse anti-FLAG M2 antibody diluted 1:500, and rabbit anti-mCherry antibody (GTX128508) diluted 1:500 in blocking solution, in a wet chamber, overnight, at 4°C. The next day, slides were washed in PBST for 10 minutes, 3 times and then incubated with goat anti-mouse Alexa 488 secondary antibody at 1:200 dilution, and goat anti-rabbit Alexa 594 secondary antibody at 1:200 dilution in PBST for 1–2 h in the dark. Slides were again washed in PBST for 10 min, 3 times and incubated with 100 µl DAPI 1 µg/ml for 5 min in the dark. Slides were finally washed in PBST in the dark for 20 min. Excess liquid was removed, and slides were mounted with Vectashield and sealed with nail polish.

## RNA extraction

For total RNA whole animal samples, young adult synchronized populations of worms were grown and washed with M9 three times to remove bacterial residues. For each 100 µl of worm pellet, 1 ml of Trizol was added. 5 cycles of freeze-cracking were carried out, freezing in liquid nitrogen, followed by thawing in a water bath at 37°C. Tubes were vortexed for 30 s on, 30 s off, for 5 min and incubated for 5 min at room temperature. 200 µl chloroform per ml of Trizol were added, tubes were shaked vigorously, incubated 2–3 min at room temperature and centrifuged at full speed for 10 min at 4°C. The top layer was transferred to a new tube, and RNA was precipitated overnight at −20°C with 1 µl glycogen and an equal volume of isopropanol.

## Germ nuclei purification and chromatin fractionation

Germ cells were purified similarly to the protocol described in Han *et al*, 2019. Approximately 50 100 mm standard NGM or RNAi plates of synchronized young adult worms were used per sample. Worms were washed off plates with M9, collected in a 15 ml conical tube and washed several times with M9 to clean from bacteria. The final wash was done with ice cold nuclear purification buffer (NPB: 10 mM HEPES pH 7.6, 10 mM KCl, 1.5 mM MgCl$_2$, 1 mM EGTA, 0.25 M sucrose, 0.025% TritonX-100, 50 mM NaF, 1 mM DTT, 40 mM β-glycerol, 2 mM Na$_3$VO$_4$). Worms were resuspended in NPB and transferred to a previously chilled metal grinder (#08-414-20A), and 1-3 strokes were applied, monitoring the extent of homogenization after each stroke, such that no more than 10-15% of worms were broken. The solution was transferred to a prechilled 50 ml conical tube, vortexed for 30 s at medium speed, and then incubated on ice for 5 min, twice. The sample was then filtered through a 100 µm nylon mesh filter (Falcon), followed by three 40 µm (Falcon) and three 20 µm filters (PluriSelect). The filtered solution was recovered and centrifuged at 100 g for 6 min at 4°C to pellet debris (usually a good germline yield has little or no material

pelleted in this step). The supernatant was recovered and centrifuged at 2,500 *g* for 10 min at 4°C to pellet nuclei, and the supernatant was discarded. Isolated nuclei were then subjected to chromatin fractionation as described in Nojima *et al*, 2016. Briefly, nuclei were resuspended in 67.5 µl of ice-cold NUN1 buffer (20 mM Tris–HCl pH = 7.9, 75 mM NaCl, 0.5 mM EDTA, 50% glycerol v/v, 1× cOmplete protease inhibitor cocktail), and mixed gently by pipetting. 0.6 ml of ice-cold NUN2 buffer (20 mM HEPES pH = 7.6, 300 mM NaCl, 0.2 mM EDTA, 7.5 mM MgCl$_2$, 1% v/v NP-40, 1 M urea, 1x cOmplete protease inhibitor cocktail) were added. The mix was vortexed at maximum speed and incubated on ice for 15 min, vortexing every 3–4 during the incubation to precipitate the chromatin fraction. The mix was then centrifuged at 16,000 *g* for 10 min at 4°C. The supernatant (nucleoplasm fraction) was collected, split into two aliquots of 300 and 900 µl Trizol LS were added to each aliquot and mixed by vortexing to proceed with RNA extraction. To isolate chromatin-bound RNA, the chromatin pellet was resuspended in 200 µl HSB buffer (10 mM Tris–HCl pH = 7.5, 500 mM NaCl, 10 mM MgCl$_2$) with 10 µl TURBO DNase (stock 2 U/µl) and 10 µl Superase-In RNase inhibitor (stock 20 U/µl) and incubated at 37°C for 10 min, vortexing to help resuspension and solubilization. 2 µl of proteinase K (stock 20 mg/ml) was then added, and the mix was incubated at 55°C for 10 min. 600 µl Trizol LS was added to proceed with RNA extraction.

## CIP-Rpph treatment of RNA and small RNA library preparation

An overview of the alternative cloning methods used in this work is available on Appendix Fig S8). To enrich for short-capped RNAs, we first pretreated 2 µg total RNA, or 500 ng of chromatin-bound or nucleoplasmic RNA with 2 µl Quick CIP (NEB) in a total volume of 20 µl for 90 min at 37°C. RNA was phenol-chloroform extracted, and precipitated at −20°C overnight with 1/10 vol 3 M AcNa, and 3 vol 100% EtOH and 1 µl glycogen. The next day, RNA was resuspended and treated with 7.5 units (1.5 µl) RNA 5′ pyrophosphohydrolase (NEB) for 1 h at 37°C in a total volume of 20 µl. RNA was extracted and precipitated overnight as described for CIP treatment. The resulting RNA was used as input for small RNA library preparation using the TruSeq small RNA kit (Illumina) according to the manufacturer's instructions, except for an increase in the number of PCR cycles from 11 to 15. An insert size range of 20–70 nt was gel purified, DNA was eluted from the gel in 0.3 M NaCl, and EtOH precipitated overnight. For direct 5′ P libraries, 1 µg total RNA was used as input, and an insert size range of 20–35 nt was gel purified. Libraries were quantitated using Qubit and Tapestation prior to pooling in groups of 6 to 12 per lane and sequenced on an Illumina HiSeq2000 (50bp single end libraries), or an Illumina NextSeq instrument (75 bp single end libraries).

## Small RNA sequencing data processing, mapping and normalization

The Illumina universal adapter was trimmed from small RNA reads using cut adapt v1.10 and reads > 17nt were mapped to WS252 *C. elegans* genome (ce11) using bowtie v0.12 (Langmead *et al*, 2009) with parameters –v 0 –m 1. Sam files were converted to bam and bed using samtools v1.2 (Li *et al*, 2009) and bedtools v2.25.0 (Quinlan & Hall, 2010).

piRNA and miRNA counts were derived from direct 5′ P libraries by searching collapsed fasta files for exact matches to existing piRNA (Batista *et al*, 2008; Gu *et al*, 2012) and miRNA (Kozomara & Griffiths-Jones, 2014) annotations. Direct library sizes were estimated, using total non-structural read counts (i), using total miRNA counts (ii) and extracting DEseq2 size factors from a miRNA counts table (iii), resulting in similar trends. Log2 fold change in abundance was calculated for loci with more than 5 DEseq2 normalized read counts on average. 1 pseudocount was added to avoid zero counts. Additionally, the change in total mature piRNA counts relative to N2 was estimated using the same size factors.

piRNA precursor reads were identified from CIP-Rpph libraries by identifying reads mapping exactly 2 nt upstream of annotated piRNAs. Library sizes were estimated as the total counts of precursor reads mapping to previously annotated TSSs (Chen *et al*, 2013). The log2 fold change in abundance was calculated for loci with more than 5 reads per million of TSS-mapping scRNA reads on average. 1 pseudocount was added to avoid zero counts. Additionally, the change in total piRNA precursor counts relative to N2 was estimated using the same size factors. To estimate the effects of the termination signal strength on piRNA precursors, we calculated termination signature strength as described in Beltran *et al*, 2019 considering the entire transcriptional unit from the TSS to +50, and considering the region downstream of 21U-RNAs alone (+25 to +50). We plotted the length distribution of precursors across equal-sized bins of piRNA loci stratified according to pausing signature strength. We additionally plotted the length distributions for the top vs bottom 20% loci.

To analyse the data on a locus-by-locus basis, we calculated the median precursor length in each locus in two alternative ways: (i) the median/mean length of all unique sequences and (ii) the median/mean length of all precursor reads. We then calculated the length shift distributions between each condition and the EV controls. We also plotted the length distributions of precursors on a locus-by-locus basis as a heat map, where each precursor sequence length for each piRNA TSS is assigned a 1 (detected) or a 0 (undetected). Additionally, we stratified these heat maps in bins of loci of increasing termination signature strength.

To assess the possibility of precursor trimming, we classified loci in bins according to the proportion of long (> 38 nt) chromatin-bound precursors observed. For each locus, we calculated the median length of all nucleoplasmic precursor sequences, and of all nucleoplasmic precursor reads. We plotted the distributions of median lengths in each of the previously defined bins as violin plots/boxplots.

### Bootstrap analysis of piRNA precursor peak locations

In order to obtain robust estimates of precursor distributions, and sufficient sampling of precursors, we combined precursors identified in the two replicates for each condition. For each condition, we generated 2,000 random subsamples of 3,000 precursor sequences, sampling without replacement and weighting the probability of sampling by the number of reads for each precursor sequence. We fit a smoothing density curve to the distribution, identified the two local maxima of the fit corresponding to first and second peaks and extracted the positions of the two. The resulting distributions of positions of both peaks for each condition are shown on Fig 3F and G.

### Degradation fragment analysis

Degradation fragment analysis was carried out from direct 5′ P small RNA libraries generated from RNA purified from germ nuclei, with the data processed as described above but without applying a minimum read length cut-off. Reads > 15 nt initiating exactly at annotated 21U-RNA sites were discarded from the libraries, and the read coverage of the remaining 5′ P unique sequences around piRNA promoters was plotted at single-nucleotide resolution in the form of an average profile and a heat map. Additionally, the counts per million and sequences per million of 5′ and 3′ ends of 5′ P reads was similarly plotted. We found an enrichment of 21-U RNA sequences initiating within a +/-5 nt window from 21U-RNA sites. These likely represent unannotated 21U-RNAs originating from overlapping TSSs from the same piRNA locus (Billi *et al*, 2013). We additionally removed these 21U-RNA sequences, as well as reads > 15 nt initiating at those unannotated 21U-RNA sites. The remaining first nucleotide and length distributions showed an enrichment of unique sequences initiating at the −2 position from annotated 21U-RNAs. Fragments whose 5′ ends mapped in between +25 and +50 from the 5′ U of annotated 21U-RNAs, corresponding to the second peak centred at +50 were collected, and their length distribution was plotted. Total fragment counts per locus were calculated and normalized to cpm of total non-structural mapped reads. The total number of loci with detectable fragments, and well as the distribution of fragment cpms were quantified across piRNA loci binned according to the strength of their termination signals.

### SLIC-CAGE library preparation and sequencing

CAGE library preparation from chromatin-bound RNA was performed using the SLiC-CAGE protocol (Cvetesic *et al*, 2018, 2019). The prepared libraries were pooled and sequenced on an Illumina HiSeq2000 (100bp paired-end libraries). In order to assess the enrichment of potential piRNA precursor reads in chromatin-bound samples, a total RNA sample control was used. Intron retention ratios were estimated using IRFinder (Middleton *et al*, 2017), which computes a ratio between the abundance of spliced and unspliced introns for all annotated RNAs, confirming the enrichment of unspliced transcripts in nascent RNA samples.

### Analysis of SLIC-CAGE data

CAGE tags of read 1 were mapped to a reference *C. elegans* genome (ce11) using STAR (Dobin *et al*, 2013). Uniquely mapped reads were imported into R (http://www.R-project.org/) as bam files using the standard workflow within the CAGEr package (Haberle *et al*, 2015). Due to the template-free activity of the reverse transcriptase enzyme, often a G is added at the 5' end of the read. G's that do not map to the genome at the 5′ end of reads are removed by CAGEr's standard workflow. G's that map to the genome are removed at a rate comparable to the mismatching G's removed. The 5′ ends of reads represent CAGE-supported transcription start sites (CTSSs), and the number of tags for each CTSS reflects expression levels. Raw tags were normalized using a referent power-law distribution and expressed as normalized tags per million (TPMs) (Balwierz *et al*, 2009).

Biological replicates were highly correlated ($r^2 > 0.9$) and were therefore merged, and the distributions of $\log_2$ fold change in CAGE

signal at the −2 position of piRNA loci were calculated. For each pair of samples, piRNA loci with non-zero counts in at least one of the conditions were included, and 0.01 tpm were added to all data points to prevent division by zero. Log$_2$ transformed data for all positions upstream of piRNAs in all conditions were plotted as a heat map, with the total sum of normalized CAGE tpms by position shown above.

For fragment length quantification, reads were aligned using using Tophat v2.1.0. (Langmead & Salzberg, 2012) with parameters -X 20000. Alignment files were sorted by read name and converted to bed12 format using pairedBamToBed12. Read pairs initiating at −2 from 21U-RNAs were retrieved, and their overall length and abundance normalized to counts per million of mapped fragments were plotted. In addition, library sizes obtained from the total numbers of pair1 reads containing SL1 and SL2 spliced leader sequences were used for normalization, resulting in very similar trends to previous normalization strategies. Tophat alignments were converted to bigwig tracks using deeptools v3.1.2 bamCoverage (Ramírez et al, 2016) with parameters -bs 10 -e, tracks from the two replicates for each condition were averaged, and signal average profiles and heat maps were generated for snRNA and protein-coding gene TSSs.

CAGE counts were randomly down sampled to multiple total count sizes (from 50,000 to 5,000,000 to match library size across libraries), and to multiple percentages of total library size (to examine saturation of piRNA detection with sequencing depth). 15 samples were obtained for each size and library, and the total number of piRNA loci with detected signal at −2 nt from the 21U-RNA were recovered. For each of the 15 subsamples, the two replicates of each library were averaged. The resulting distributions of total detected piRNAs per condition were plotted as boxplots against the depth of the subsamples.

### ATAC-seq and ChIP-seq data processing and analysis

Processed bigwig ATAC-seq tracks from young adult wild-type nematodes, and day 1 glp-1 mutant adults were obtained from Janes et al (2018) (GSE114439) and lifted over to ce11. Raw H3K27 acetylation ChIP-seq data from isolated germ nuclei and somatic nuclei were obtained from Han et al, 2019 (SRR9214969 to SRR9214976), mapped to ce11 using bowtie2 (Langmead & Salzberg, 2012) with default parameters, and the resulting bam files were converted to bigwig tracks using deeptools bamCoverage (Ramírez et al, 2016) with parameters -bs 1 -e 150 –normalize Using RPGC --effectiveGenomeSize 96945445. Replicates were averaged using bigwigCompare, and heat maps of H3K27ac and ATAC signal around piRNA loci were generated using deeptools computeMatrix reference-point. The signal data underlying the heat maps were extracted in order to compare the distributions of ATAC (±100 bp around piRNA TSSs) and H3K27ac signal between groups of loci (±1 kb around piRNA TSSs).

### Differential expression analysis of gene promoters genome-wide using scRNA and CAGE count data

scRNA or CAGE TSS annotations were intersected with a set of WormBase protein-coding gene promoters (Chen et al, 2013) lifted over to ce11, in order to generate count matrices of scRNA or CAGE counts for all gene promoters genome-wide. Differential expression analysis comparing N2 empty vector chromatin-bound samples to N2 ints-11 RNAi chromatin-bound samples was carried out using DESeq2. Promoters were considered to be differentially expressed using a Benjamini-Hochberg FDR threshold of 0.1. The fold changes of promoters identified as differentially expressed in either of the two experiments were extracted, and the fold change in scRNA signal was plotted against the fold change in CAGE signal. To test for enrichment of motif-independent piRNAs in each of the promoter classes, motif-independent piRNAs were first overlapped with promoters detected in the scRNA and CAGE experiments in order to determine their background enrichment. Enrichment was calculated in each promoter set as an odds ratio relative to its corresponding background; for instance the fraction of CAGE "up" promoters overlapping with motif-independent piRNAs was compared with the fraction of all CAGE detectable promoters overlapping with motif-independent piRNAs. For intersecting promoter sets, genes detected as significant in either experiment, and having a fold change greater than 2 in both experiments were selected. In these cases, background was calculated as the fraction of promoters detected in both experiments overlapping with motif-independent piRNAs.

To investigate transcriptional changes in transposable element (TE) expression, we used RepeatMasker and RepeatModeller to identify the positions of TEs within the C. elegans genome, version ce11. We then used Bedtools to intersect this with the normalized CAGE data resulting in counts for each transposable element. The maximum CAGE signal across the two replicates was identified and the sum of this for TE families compared between EV-treated animals and int-11 RNAi-treated animals using a Wilcoxon paired test across individual elements of each family. Significantly different TEs were identified as those with a Benjamani-Hochberg adjusted P-value < 0.05 and a more than 1.5-fold difference in total expression.

## Data availability

Sequencing data (small RNA sequencing, short-capped RNA sequencing and CAGE data): Sequence read archive (SRA) PRJNA592867 (https://www.ncbi.nlm.nih.gov/bioproject/PRJNA592867).

**Expanded View** for this article is available online.

### Acknowledgements

We would like to thank Juan Cabello for kindly sharing JCP383 ints-6::GFP nematodes with us at the beginning of the study. This work was funded by the United Kingdom Medical Research Council (MC-A652-5PY80 "Epigenetics and Evolution"-PS; MC_UP_1102/1 "Computational Regulatory Genomics"-BL ) and by the Wellcome Trust (106954/Z/15/Z- BL). PS is a member of the EMBO Young Investigator scheme.

### Author contributions

TB and PS conceived the project and designed the experiments. TB performed all C. elegans experiments with assistance from SG. EP performed CAGE experiments. TB performed most computational analyses with some supplemental analyses performed by PS and, for CAGE analyses EP and BL. TB and PS wrote the first draft. TB, EP, SG, BL and PS participated in review and editing.

## Conflict of interest

The authors declare that they have no conflict of interest.

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
