## [Review Process File · The EMBO Journal]

Integrator is recruited to promoter-proximally paused RNA Pol II to generate *C. elegans* piRNA precursors

Toni Beltran, Eleni Pahita, Subhanita Ghosh, Boris Lenhard, and Peter Sarkies
DOI: [10.15252/embj.2020105564](https://doi.org/10.15252/embj.2020105564)

Corresponding author(s): Peter Sarkies (psarkies@imperial.ac.uk)

Review Timeline:

Submission Date:	7th May 20
Editorial Decision:	9th Jun 20
Revision Received:	2nd Sep 20
Editorial Decision:	29th Sep 20
Revision Received:	14th Oct 20
Accepted:	27th Oct 20

Editor: Stefanie Boehm

Transaction Report:

Thank you for submitting your manuscript for consideration by The EMBO Journal. Please also excuse the delay in communicating the decision to you, which was due to a delayed review process on account of the current pandemic. We have now however received three referee reports on your study, which are included below for your information.

As you will see, the reviewers are overall positive and acknowledge the interest of the field in the topic and the study. Nonetheless they also raise some concerns that would need to be addressed in a revised manuscript. In particular, the referees find that additional analyses on the piRNA precursors affected by Integrator loss should be performed (ref#1 major point 1; ref#2 major points 1, 3) as well as addressing if potential signatures of motifs exist (ref#1 major point 5; ref#3). In addition, the points referee #1 (2-4) and referee #2 (major 2) raise should be addressed by additional controls, reanalysis of acquired data and/or appropriate discussion. Please also carefully consider and respond to all other comments the referees have and revise the manuscript text accordingly.

Please note that it is our policy to allow only a single round of major revision. We realize that lab work worldwide is currently affected by the COVID-19/SARS-CoV-2 pandemic and that an experimental revision may be delayed. We can extend the revision time when needed, and we have extended our 'scooping protection policy' to cover the period required for a full revision. However, it is nonetheless important to clarify any questions and concerns at this stage and we encourage you to discuss a revision plan and any potential issues you may foresee as soon as possible.

Please also feel free to contact me should you have any other further questions. Thank you for the opportunity to consider your work for publication. I look forward to receiving your revised manuscript.

Referee #1:

This manuscript reports the requirement of the Integrator complex in the production of a subset of piRNA precursors in *C. elegans*. The authors revealed two distinct populations of nascent capped piRNA precursors, ~28 nt and ~48 nt, representing Pol II pause sites on the chromatin, and proposed that Integrator cleaves ~48 nt piRNA precursors into shorter ~38 nt fragments, promoting generation of mature piRNAs. Moreover, TFIIS, which rescues Pol II backtracking, is important for Integrator-mediated processing of piRNA precursors by promoting Pol II transcription through the +48 pause site. Overall, the proposed model is interesting and beneficial to the field, highlighting the similarity (and difference) between piRNA and snRNA biogenesis pathways in worms.

Major Points:

1. It is proposed that Integrator cleaves ~48 nt piRNA precursors into ~38 nt. However, the differences in the overall abundance (Figure 1A, B) and peak distributions as mixtures (Figure 1C, D and 3) are rather superficial and weak as evidence. The authors should perform more rigorous analyses by separately tracing the processing pathway of each piRNA precursor sequence (from each distinguishable piRNA locus) and comprehensively integrating them after appropriate categorization (motifs, AT-richness etc.).
2. In Figure 3D, it appears that the abundance of extended ~29-30 nt precursors in *tfiis* ints-11 RNAi is markedly lower than that in *tfiis* EV. How do the authors interpret this effect?
3. Can the authors rule out the possibility that, in addition to the proposed cleavage of ~48 nt into ~38 nt precursors, Integrator promotes Pol II pausing at the ~28 nt peak position?
4. What is the fate of the ~38 nt piRNA precursors? Are they further processed into ~28 nt precursors before matured into 21U-RNAs?
5. Are there any signatures/motifs that account for pol II pausing at the ~48 nt sites?

Minor Points:

1. Fig. S3B is not cited in the text.
2. Page 10, line 5: "Domingues et al." should probably read "Rodrigues et al."
3. There is no legend for Fig. S1E.
4. The analysis in Fig. S4C should be extended to *tfiis*/ints-11 RNAi.
5. Additional clarification is required for the X-axis numbers in Fig. S4D etc.
6. The Y-axis of Fig. S6B is neither linear nor log-scale.
7. The direction of the X-axis should be reversed in Fig. S6C-E.
8. Fitting is rather irrelevant for Fig. S8G.
9. Fig. 6 is somewhat misleading, because it appears as if all the piRNA precursors are generated in a motif-dependent manner.

Referee #2:

The manuscript by Beltran et al presents the identification of the Integrator RNA cleavage complex as an important factor in *c. elegans* piRNA biogenesis. This finding is supported by observations of decreased piRNA production and piRNA silencing activity in RNAi-depletion of the Integrator component Ints-11. Further support is lent by a striking subcellular colocalization of Integrator subunits with piRNA factors as well as analyses of transcript phenotypes by high-throughput sequencing.

The role of Integrator in *c. elegans* piRNA biology is highly interesting and will attract the attention of a broad readership on both piRNA biology and co-transcriptional processing. The study is comprehensive and for the large part carefully analyzed (with a few notable exceptions listed below). My main critique is that the analyses and interpretation of the transcript patterns at piRNA loci lacks clarity in important places. I believe this can be addressed mainly computationally as

requested below. Further experimental evidence would be very interesting, but is non-essential for the present publication.

Major concerns

1. The authors show that ints-11 RNAi depletes 21U piRNA. What are the precursors for the missing integrator-dependent piRNAs? According to the authors' model (Fig 6) the main class of +28 nt precursors are Integrator-independent. The observed peak of precursor 3' ends at +48 nt is not consistent with processing from the Integrator-dependent cleavage at +38 nt (Fig 4B). This question should be clarified in text, analyses or experimentally.

2. Missing data on transcript 3' ends: Page 7, line 8 reads: "3' ends centered 58 nt downstream of piRNA TSSs (Supp. Fig. 4C)". However, Figure S4C seems to be swapped with S4B and S4B looks identical to main figure 4B. It therefore seems likely that the mentioned plot for 3' ends is missing. The authors should clarify and correct this.

3. The plotting of relative fractions of sequences in Figure 3B and 3E are difficult to interpret. Given the overall decrease in precursor levels in tffis ints-11 RNAi (Fig 3D), the shifts to longer precursors in Fig 3E could still be consistent with a scenario where, in absolute numbers, fewer long precursors are produced in knock-down/mutants. It thus seems plausible that not only elongation is affected, but that transcription initiation levels and/or transcript stability (for tffis where nucleoplasmic levels are down, but chr-ass. levels are unchanged). These possibilities should be addressed in the manuscript. Analyses of RNA exosome RNAi-depletion in the tffis background could potentially address this latter option.

4. The panel focus for main Figures 1 and 5 is not well aligned with the key points emphasized in the text. Specifically:

a. For Figure 1, one key message is "We show that Integrator activity is required for accumulation of mature piRNAs and their silencing activity." [page 3]. The data that most clearly shows the loss of piRNAs is Fig S1C and for the silencing effect the data is in Fig S1D. Rather than focusing on replicate reproducibility (current Fig 1A-B) it would increase readability to shift the key data (S1C-D) to the main figure 1.

b. For Figure 5, the central message is that "Pol II elongation control is independent of Integrator activity at the majority of piRNA loci" (page 7), yet the related main figure 5 focuses almost entirely on the few percent of motif-dependent piRNA loci that do produce read-through. Just looking at the presented data, the reader therefore may get the opposite impression from the reading the text. This could be improved by including all piRNA loci in the analysis to make it evident that the read-through cases are outliers.

Minor concerns

5. A number of different RNA cloning methods are used throughout the manuscript. To help figure readability, it would be good to include an indication of the library cloning selection steps. This could be either as a schematic in connection with the relevant panel or a more detailed annotation on the panels.

6. "EV" as used through-out the paper has not been defined (assuming EV=empty vector)

7. Page 8: two numbers are stated for the fraction of motif-dependent piRNA loci that give rise to read-through in ints-11 knock-down, 3 % (line 7) and 4.65 % (line 27). It is unclear why these numbers are different and which population they refer to.

8. Page 12, line 27: Handler et al is a somatic screen (where they also found Integrator!). The correct reference for the germline screen is Czech et al 2013

9. The color coding could be improved to increase figure readability. Within figure 3, for example, orange indicates, PETISCO complex, chromatin-bound, weak terminator and N2 EV.

10. Given the complexity of the transcriptional regulation analyses it is important to ensure high clarity in the wording. The manuscript should be edited for clarity with this in mind; here just one example: "The observation that Integrator processing occurs downstream of the +48 pause site .." As I understand the authors model, Integrator processing occurs upstream of the +48 pause site (around +38), but with RNAPII having progressed to downstream of the pause site. This is not clear from the sentence.

11. The figure legends often lack a description of the unit of the plotted data. E.g. Fig. 3C: counts of what? And Fig. 3E: fraction of which sequences? This should be added.

12. Figure 3E: the logic behind statement Page6, Line 18-19 is unclear: "This shift persisted upon ints-11 knockdown in a tfiis mutant background, indicating that it is not Integrator-dependent (Figure 3E,F,G)". If the cleavage would be Integrator-dependent then it should be observed in ints-11 RNAi alone. Alternatively, a model would require Integrator to be an inhibitor of TFIIIS-activated RNAPII cleavage, which seems unmotivated. Please clarify.

13. Beyond the introduction, the authors do not mention regulation of transposable elements. Given the literature context of the piRNA pathway, it would be good to briefly comment on the absence of transposon deregulation (my assumption) in tfiis and ints-11 perturbation.

Referee #3:

This paper contains important, new data on piRNA biogenesis in *C. elegans*. Previous studies suggest that piRNA promoters evolved from snRNA promoters in *C. elegans*. Thus, the authors explore the parallels between the mechanisms of piRNA and snRNA transcription. They show that the catalytic activity of Integrator complex, which terminates snRNA transcription, cleaves nascent capped piRNA precursors associated with promoter-proximal Pol II, resulting termination of transcription. However Integrator depletion did not result in readthrough transcription at most motif-dependent piRNA loci indicating that the cleavage activity of Integrator is not required to prevent elongation at motif-dependent piRNAs. Thus, Pol II elongation control is likely to operate independently of Integrator via additional mechanisms. Experiments are well designed with appropriate controls. The paper is well written and concise, and the data are presented clearly. Studies of this sort are good references and resources for further comparisons.

The authors found that motif-dependent piRNAs localize to broad domains of H3K27me3, which also have low accessibility. They speculate that the tight control of Pol II elongation at motif-dependent piRNA loci may be a consequence of evolutionary pressure to avoid transcriptional interference between neighbouring piRNA loci given the high density of loci in piRNA cluster regions that are simultaneously activated. I wonder if there exist some specific sequence features in the inter-piRNA loci (regions between piRNA loci). It is known that in gene clusters in mammals such as olfactory receptor gene clusters, transposons such as L1 elements intervene between protein-coding genes (e.g., Kambere & Lane, *J Mol Evol* 2009).

Referee #1:

This manuscript reports the requirement of the Integrator complex in the production of a subset of piRNA precursors in *C. elegans*. The authors revealed two distinct populations of nascent capped piRNA precursors, ~28 nt and ~48 nt, representing Pol II pause sites on the chromatin, and proposed that Integrator cleaves ~48 nt piRNA precursors into shorter ~38 nt fragments, promoting generation of mature piRNAs. Moreover, TFIIS, which rescues Pol II backtracking, is important for Integrator-mediated processing of piRNA precursors by promoting Pol II transcription through the +48 pause site. Overall, the proposed model is interesting and beneficial to the field, highlighting the similarity (and difference) between piRNA and snRNA biogenesis pathways in worms.

Major Points:

1. It is proposed that Integrator cleaves ~48 nt piRNA precursors into ~38 nt. However, the differences in the overall abundance (Figure 1A, B) and peak distributions as mixtures (Figure 1C, D and 3) are rather superficial and weak as evidence. The authors should perform more rigorous analyses by separately tracing the processing pathway of each piRNA precursor sequence (from each distinguishable piRNA locus) and comprehensively integrating them after appropriate categorization (motifs, AT-richness etc.).

We thank the reviewer for this suggestion. To support the conclusions that we made from pooling piRNAs together we have added additional figures to analyse each locus individually throughout the manuscript. In Figure 1 we now show changes in total piRNA levels (as suggested by Reviewer 2) and locus-by-locus comparisons. In addition, we have added an analysis of precursor length on a locus-by-locus basis in Supp. Fig. 1. These data recapitulate the trends seen in the overall distributions.

“The increase in precursor length observed from the overall distributions was confirmed by comparing the difference in length at each locus between *ints-11* RNAi and EV controls (Supp. Fig 1G,H)”

To complement Figure 3 we now add two revised Supplementary Figures. In Supplementary Figure 4, we analyse the length changes in nucleoplasmic and chromatin-bound RNAs across genotypes and RNAi treatments on a locus-by-locus basis. In addition, in Supplementary Figure 5, we show how the strength of the motif influences the length of chromatin bound precursors detected at a locus-by-locus level across the different conditions. This shows that both long and short chromatin bound RNA can be detected at many loci thus the average profile is a good representation of what happens at individual loci. Further, this provides a clear confirmation of our result that loci with weaker termination signals demonstrate more evidence of longer precursors in wild type and show that integrator RNAi reduces the relative abundance of shorter precursors across the range of termination signal strength, supporting the conclusions from aggregated piRNA loci.

We have now referred to these new additions in the results section:

“To support these observations at a locus-specific level, we calculated the median length changes for all loci (Supp. Fig. 4A) and plotted the lengths of all detectable precursor sequences in each locus as a heatmap (Supp fig 4B). In addition, we

plotted chromatin-bound precursor length heatmaps across bins of increasing pausing strength (Supp. Fig. 5). In EV controls, loci with weak pausing signals generally had both ~20-30 nt and ~40-50 nt long chromatin bound RNAs, indicating that both types of pausing events could occur at the same locus. However, at loci with strong pausing signals, a larger proportion of loci exhibited only ~20-30 nt chromatin-bound precursors. Similar trends were observed in ints-11 RNAi-treated wild-type animals, and in *tfiis* mutants.”

2. In Figure 3D, it appears that the abundance of extended ~29-30 nt precursors in *tfiis* ints-11 RNAi is markedly lower than that in *tfiis* EV. How do the authors interpret this effect?

We thank the reviewer for this observation. Our model is that the reduced efficiency of termination, most marked in the absence of both TFIIIS and integrator activity means that RNA polIII is not released at the 3' end hence increased accumulation at the second peak (48nt). We predict that due to spatial constraints it is not possible for RNA polymerase II to initiate at the piRNA promoter when there is an RNA polymerase II is still bound at the 3' end. This would lead to decreased accumulation at the 28nt peak. However, we do not know for certain that two polymerases cannot be loaded at the same locus simultaneously thus this explanation is still speculative.

“The peak at 28 nt was also reduced relative to *tfiis* mutants, suggesting that the combined effect of loss of TFIIIS and reduced Integrator activity interferes with initiation of transcription. This may result from torsional stress or spatial constraints resulting from the accumulation of Pol II at the 3' end of the piRNA locus.”

3. Can the authors rule out the possibility that, in addition to the proposed cleavage of ~48 nt into ~38 nt precursors, Integrator promotes Pol II pausing at the ~28 nt peak position?

This is an interesting idea but given the known function of integrator at snRNA loci, it seems unlikely that it would also promote pausing at the 28nt position. However, we cannot rule it out formally- we have now acknowledged this point in the discussion section.

“Of note, we cannot formally rule out the possibility that Integrator promotes accumulation of Pol II at the +28 site, although we favour the model outlined above given the known functions of Integrator in nascent RNA cleavage.”

4. What is the fate of the ~38 nt piRNA precursors? Are they further processed into ~28 nt precursors before matured into 21U-RNAs?

To address this point we now add plots showing the correlation between the median length of chromatin-bound RNA with the median length of nucleoplasmic RNA derived from the same locus. This shows that longer piRNA precursors on chromatin nevertheless correspond to accumulation of 28nt precursors, which suggests that 38nt piRNA precursors released by integrator cleavage may be further trimmed (supplemental figure 8).

“We wondered whether we could find any evidence of additional trimming of Integrator-released precursors. To test this, we categorized loci in bins according to the median length of their chromatin-bound precursors, and examined the

distributions of nucleoplasmic precursor length across these bins (Supp. Fig. 8). This analysis showed that even loci with a large fraction of long chromatin-bound precursors (>38 nt) produce nucleoplasmic precursors with a median length <30 nt (Supp. Fig 8). This implies that RNA released from chromatin upon Integrator cleavage may be further trimmed to 28 nt in the nucleoplasm before further maturation to mature piRNAs.“

5. Are there any signatures/motifs that account for pol II pausing at the ~48 nt sites?

We did not find any enriched sequence motifs downstream of the 48nt sites. However, we analysed MNase seq data (Jeffers et al., 2017) which shows a small enrichment of MNase-seq data centered around +125nt from the TSS. Assuming the nucleosome wraps 146bp, one can subtract half a nucleosome (73nt) from 125, which corresponds to +52nt from the TSS, almost exactly matching the second pause site (see Figure below). This would be consistent with a recent study (Aoi et al., 2020) where two pause sites on protein-coding gene promoters are described, with the second site coinciding with positioned nucleosomes as measured by MNase-seq. We however note that the causal relationship between this nucleosome and transcription is unclear, and as a result we have not included this in the manuscript.

Minor Points:

1. Fig. S3B is not cited in the text.

Done

2. Page 10, line 5: "Domingues et al." should probably read "Rodrigues et al."

Done

3. There is no legend for Fig. S1E.

Done

4. The analysis in Fig. S4C should be extended to tfiis/int-11 RNAi.

Done

5. Additional clarification is required for the X-axis numbers in Fig. S4D etc.

Done

6. The Y-axis of Fig. S6B is neither linear nor log-scale.

In this axis one can better appreciate the differences between samples as most of the data is contained within 0-0.25.

7. The direction of the X-axis should be reversed in Fig. S6C-E.

Done

8. Fitting is rather irrelevant for Fig. S8G.

We have removed the fitted line.

9. Fig. 6 is somewhat misleading, because it appears as if all the piRNA precursors are generated in a motif-dependent manner.

We have now clarified this by including a "motif-dependent" label.

Referee #2:

The manuscript by Beltran et al presents the identification of the Integrator RNA cleavage complex as an important factor in *c. elegans* piRNA biogenesis. This finding is supported by observations of decreased piRNA production and piRNA silencing activity in RNAi-depletion of the Integrator component Ints-11. Further support is lent by a striking subcellular colocalization of Integrator subunits with piRNA factors as well as analyses of transcript phenotypes by high-throughput sequencing.

The role of Integrator in *c. elegans* piRNA biology is highly interesting and will attract the attention of a broad readership on both piRNA biology and co-transcriptional processing. The study is comprehensive and for the large part carefully analyzed (with a few notable exceptions listed below). My main critique is that the analyses and interpretation of the transcript patterns at piRNA loci lacks clarity in important places. I believe this can be addressed mainly computationally as requested below. Further experimental evidence would be very interesting, but is non-essential for the present publication.

Major concerns

1. The authors show that int-11 RNAi depletes 21U piRNA. What are the precursors for the missing integrator-dependent piRNAs? According to the authors' model (Fig 6) the main class of +28 nt precursors are Integrator-independent. The observed peak of precursor 3' ends at +48 nt is not consistent with processing from the Integrator-dependent cleavage at +38 nt (Fig 4B). This question should be clarified in text, analyses or experimentally.

We apologise for any confusion here. The peak we observe at 48nt is on chromatin-bound RNA thus represents the predominant position of the RNA polymerase rather than the position of cleavage. By mapping the 5' position of putative cleavage products we show that

the actual cleavage site is at 38nt rather than 48nt, consistent with Integrator cleaving upstream of the position of the polymerase. Furthermore, the 3' end of these fragments maps to ~58nt downstream of the TSS, which means that the polymerase pauses at 48nt but this pause event must be resolved and the polymerase transcribes beyond the pause site before cleavage can occur. We have expanded the explanation of this chain of events in the results section.

2. Missing data on transcript 3' ends: Page 7, line 8 reads: "3' ends centered 58 nt downstream of piRNA TSSs (Supp. Fig. 4C)". However, Figure S4C seems to be swapped with S4B and S4B looks identical to main figure 4B. It therefore seems likely that the mentioned plot for 3' ends is missing. The authors should clarify and correct this.

We thank the reviewer for pointing this out – we have now included the 3' end positions for all conditions as suggested by Reviewer 1 in Supp. Fig 6C.

3. The plotting of relative fractions of sequences in Figure 3B and 3E are difficult to interpret. Given the overall decrease in precursor levels in *tfiis* ints-11 RNAi (Fig 3D), the shifts to longer precursors in Fig 3E could still be consistent with a scenario where, in absolute numbers, fewer long precursors are produced in knock-down/mutants. It thus seems plausible that not only elongation is affected, but that transcription initiation levels and/or transcript stability (for *tfiis* where nucleoplasmic levels are down, but chr-ass. levels are unchanged). These possibilities should be addressed in the manuscript. Analyses of RNA exosome RNAi-depletion in the *tfiis* background could potentially address this latter option.

We agree with the reviewer that there is a possibility that transcription initiation is affected in ints-11 RNAi. This may be a secondary effect caused by the defect in cleavage and release of the polymerase leading to a “molecular traffic jam” at the piRNA loci. We have addressed this in the manuscript as follows (see also response to point 2 raised by Reviewer 1 above).

“The peak at 28 nt was also reduced relative to *tfiis* mutants, suggesting that the combined effect of loss of TFIIIS and reduced Integrator activity interferes with initiation of transcription. This may result from torsional stress or spatial constraints resulting from the accumulation of Pol II at the 3' end of the piRNA locus.”

We have also emphasized in the discussion that Integrator may affect transcription initiation and/or stability:

It is also possible that Integrator depletion affects overall transcription initiation or nascent piRNA precursor stability, given the decrease in the levels of chromatin-bound precursors observed upon *ints-11* RNAi.

4. The panel focus for main Figures 1 and 5 is not well aligned with the key points emphasized in the text. Specifically:

a. For Figure 1, one key message is "We show that Integrator activity is required for accumulation of mature piRNAs and their silencing activity. " [page 3]. The data that most clearly shows the loss of piRNAs is Fig S1C and for the silencing effect the data is in Fig

S1D. Rather than focusing on replicate reproducibility (current Fig 1A-B) it would increase readability to shift the key data (S1C-D) to the main figure 1.

We thank the reviewer for this suggestion, which we have implemented in the revised version.

b. For Figure 5, the central message is that "Pol II elongation control is independent of Integrator activity at the majority of piRNA loci" (page 7), yet the related main figure 5 focuses almost entirely on the few percent of motif-dependent piRNA loci that do produce read-through. Just looking at the presented data, the reader therefore may get the opposite impression from the reading the text. This could be improved by including all piRNA loci in the analysis to make it evident that the read-through cases are outliers.

We have now added a barplot to Figure 5 that shows the fraction of all piRNA loci rather than the total reads. This clarifies the point that only a minority of piRNA loci show detectable readthrough.

Minor concerns

5. A number of different RNA cloning methods are used throughout the manuscript. To help figure readability, it would be good to include an indication of the library cloning selection steps. This could be either as a schematic in connection with the relevant panel or a more detailed annotation on the panels.

We thank the reviewer for this suggestion. We have incorporated an extra supplemental figure (Supp. Fig. 13) describing the cloning methods used graphically.

6. "EV" as used through-out the paper has not been defined (assuming EV=empty vector)

We thank the reviewer for pointing this out – we have included the abbreviating in the first mention at the beginning of the results section.

7. Page 8: two numbers are stated for the fraction of motif-dependent piRNA loci that give rise to read-through in ints-11 knock-down, 3 % (line 7) and 4.65 % (line 27). It is unclear why these numbers are different and which population they refer to.

Done.

8. Page 12, line 27: Handler et al is a somatic screen (where they also found Integrator!). The correct reference for the germline screen is Czech et al 2013.

We thank the reviewer for pointing this out – we now have included both references and clarified the distinction between the two screens (germline vs ovarian somatic support cells).

9. The color coding could be improved to increase figure readability. Within figure 3, for example, orange indicates, PETISCO complex, chromatin-bound, weak terminator and N2 EV.

We have changed PETISCO to blue, chromatin to orange to match the colors in the figure.

10. Given the complexity of the transcriptional regulation analyses it is important to ensure

high clarity in the wording. The manuscript should be edited for clarity with this in mind; here just one example: "The observation that Integrator processing occurs downstream of the +48 pause site .." As I understand the authors model, Integrator processing occurs upstream of the +48 pause site (around +38), but with RNAPII having progressed to downstream of the pause site. This is not clear from the sentence.

We have edited the manuscript to clarify our model of how Pol II termination is linked to integrator cleavage (also see above).

11. The figure legends often lack a description of the unit of the plotted data. E.g. Fig. 3C: counts of what? And Fig. 3E: fraction of which sequences? This should be added.

We have clarified these in the figure axes and in the legends.

12. Figure 3E: the logic behind statement Page6, Line 18-19 is unclear: "This shift persisted upon ints-11 knockdown in a tfiis mutant background, indicating that it is not Integrator-dependent (Figure 3E,F,G)". If the cleavage would be Integrator-dependent then it should be observed in ints-11 RNAi alone. Alternatively, a model would require Integrator to be an inhibitor of TFIIS-activated RNAPII cleavage, which seems unmotivated. Please clarify.

We apologise for the lack of clarity here. These results indicate that Integrator is not required for TFIIS-activated Pol II cleavage because the shift is not present in ints-11 RNAi but still occurs when TFIIS is mutated and ints-11 is knocked down.

13. Beyond the introduction, the authors do not mention regulation of transposable elements. Given the literature context of the piRNA pathway, it would be good to briefly comment on the absence of transposon deregulation (my assumption) in tfiis and ints-11 perturbation.

We have analysed transposon expression using the CAGE data and we show that there is a small subset of Tc1 elements that show significant deregulation upon ints-11 knockdown. Given that the effect is specific for this subset, we suggest that this may indicate a role for integrator in regulating Tc1 expression through termination of transcription, similar to its role at certain protein-coding genes. This data is in Supplementary Figure 12.

Referee #3:

This paper contains important, new data on piRNA biogenesis in *C. elegans*. Previous studies suggest that piRNA promoters evolved from snRNA promoters in *C. elegans*. Thus, the authors explore the parallels between the mechanisms of piRNA and snRNA transcription. They show that the catalytic activity of Integrator complex, which terminates snRNA transcription, cleaves nascent capped piRNA precursors associated with promoter-proximal Pol II, resulting termination of transcription. However, Integrator depletion did not result in readthrough transcription at most motif-dependent piRNA loci, indicating that the cleavage activity of Integrator is not required to prevent elongation at motif-dependent piRNAs. Thus, Pol II elongation control is likely to operate independently of Integrator via additional mechanisms. Experiments are well designed with appropriate controls. The paper is well written and concise, and the data are presented clearly. Studies of this sort are good references and resources for further comparisons.

The authors found that motif-dependent piRNAs localize to broad domains of H3K27me3, which also have low accessibility. They speculate that the tight control of Pol II elongation at motif-dependent piRNA loci may be a consequence of evolutionary pressure to avoid transcriptional interference between neighbouring piRNA loci given the high density of loci in piRNA cluster regions that are simultaneously activated. I wonder if there exist some specific sequence features in the inter-piRNA loci (regions between piRNA loci). It is known that in gene clusters in mammals such as olfactory receptor gene clusters, transposons such as L1 elements intervene between protein-coding genes (e.g., Kambere & Lane, J Mol Evol 2009).

Contrary to piRNA clusters in *Drosophila*, piRNAs in *C. elegans* are not particularly dense in transposable elements or other repetitive sequences. Instead, they are located in genomic regions that contain developmentally regulated protein-coding genes that are expressed in specific tissues but silenced in the germline. piRNAs localize to small intergenic regions in between those genes, and can sometimes be intronic as well (although there is a bias towards intergenic regions) (Beltran et al., 2019). The density of piRNAs in these regions is extremely high, such that there are only 100-300 intervening bp in between loci. We have also examined other nematode species, and in some other nematodes more distantly related to *C. elegans*, such as *P. pacificus*, piRNAs are actually expressed from the introns of active protein-coding genes (Beltran et al., 2019). So it seems that these loci can function in a variety of genomic contexts in different species, even though within a species their location in specific chromatin domains and regions is essential for expression, as we showed using CRISPR-mediated genome editing in the case of *C. elegans* piRNAs (Beltran et al., 2019).

We have also tried to determine whether any specific sequence features explain the second pause site (+48), and although we did not find any enriched motifs, we found suggestive evidence of a positioned nucleosome downstream of this pause site (see response to Reviewer 1).

Thank you for submitting your revised manuscript, we have now received the reports from the three initial referees (see comments below). I am pleased to say that they overall find that their comments have been satisfactorily addressed and now support publication. Referee #1 raises some remaining issues regarding figure labelling and references to figures in the text, please carefully check these points and revise the manuscript as needed. In addition, I would like to ask you to also address a number of editorial issues that are listed in detail below. Please make any changes to the manuscript text in the attached document only using the "track changes" option. Once these remaining issues are resolved, we will be happy to formally accept the manuscript for publication.

Thank you again for giving us the chance to consider your manuscript for The EMBO Journal. I look forward to receiving your final revision. Please feel free to contact me if you have further questions regarding the revision or any of the specific points listed below.

Referee #1:

The authors have adequately addressed my previous concerns. The following points need to be amended before publication.

1. Fig. 1A (looks like old Fig. S1F; now labeled as "mature") and 1C (looks like old Fig. S1C; now labeled as "precursor") might have been switched. Please double-check.
2. Fig. S1F is cited after S1G and H in the text. The order of figure panels needs to be changed.
3. Please explain "collapsed sequences" and "uncollapsed reads" in Fig. S4A. Also, "mean" should read "median" (at least according to the text).
4. Fig. S4B requires clear labeling (N2EV, N2 ints-11 RNAi etc.) in the figure.
5. In Fig. S8, the "(0.8,1]" group shows ~40 nt median length of precursors in the nucleoplasmic fraction. Are ~28 nt precursors decreased by ints-11 RNAi even in this specific group?
6. The fitted line in fig. S11G (old Fig. S8G) is remaining, even though the point-by-point response states that it was removed.
7. Fig. S12 must be cited in the text or removed.
8. Page 12, line 34-35: Fig. S10 (instead of Fig. S11) should probably be cited here.
9. Page 15, line 30: Fig. S13 (instead of Fig. S11) should probably be cited here.
10. Page 17, line 19: Fig. 4F and G, which do not exist in the current manuscript, are cited.

Referee #2:

The authors have very carefully addressed all my concerns (as well as those of the other reviewers) and I consider the manuscript ready for publication.

Referee #3:

The paper has been improved and the authors have addressed most of the reviewers' concerns. There are no further technical concerns about the content.

Referee #1:

The authors have adequately addressed my previous concerns. The following points need to be amended before publication.

1. Fig. 1A (looks like old Fig. S1F; now labeled as "mature") and 1C (looks like old Fig. S1C; now labeled as "precursor") might have been switched. Please double-check.

These were initially swapped in the first submission and were corrected during the revision.

2. Fig. S1F is cited after S1G and H in the text. The order of figure panels needs to be changed.

We have now updated the numbering in the figure to reflect the order of appearance in the text.

3. Please explain "collapsed sequences" and "uncollapsed reads" in Fig. S4A. Also, "mean" should read "median" (at least according to the text).

We have updated these to "unique detected sequences" and "all detected reads" to clarify. We plotted the mean change, so we have updated the text accordingly.

4. Fig. S4B requires clear labeling (N2EV, N2 ints-11 RNAi etc.) in the figure.

Done.

5. In Fig. S8, the "(0.8,1]" group shows ~40 nt median length of precursors in the nucleoplasmic fraction. Are ~28 nt precursors decreased by ints-11 RNAi even in this specific group?

Yes, although in some samples there is a tendency for loci with longer chromatin-bound precursors (top 20%) to have a stronger downregulation of nucleoplasmic precursors, compared to loci with shorter chromatin-bound precursors (bottom 20%).

6. The fitted line in fig. S11G (old Fig. S8G) is remaining, even though the point-by-point response states that it was removed.

Done.

7. Fig. S12 must be cited in the text or removed.

Done.

8. Page 12, line 34-35: Fig. S10 (instead of Fig. S11) should probably be cited here.

Done.

9. Page 15, line 30: Fig. S13 (instead of Fig. S11) should probably be cited here.

Done.

10. Page 17, line 19: Fig. 4F and G, which do not exist in the current manuscript, are cited.

Referee #2:

The authors have very carefully addressed all my concerns (as well as those of the other reviewers) and I consider the manuscript ready for publication.

Referee #3:

The paper has been improved and the authors have addressed most of the reviewers' concerns. There are no further technical concerns about the content.

Thank you again for submitting the final revised version of your manuscript for our consideration. I am pleased to inform you that we have now accepted it for publication in The EMBO Journal.

Corresponding Author Name: Peter SARKIES

Journal Submitted to: The EMBO Journal

Manuscript Number: EMBOJ-2020-105564